



# Long-term total OH reactivity measurements in a boreal forest

Arnaud P. Praplan[1], Toni Tykkä[1], Dean Chen[2], Michael Boy[2], Ditte Taipale[2], Ville Vakkari[1,3], Putian Zhou[2], Tuukka Petäjä[2], and Heidi Hellén[1]

[1]Finnish Meteorological Institute, P.O. Box 503, 00101 Helsinki, Finland
[2]Institute for Atmospheric and Earth System Research/Physics, Faculty of Science, P.O. Box 64, 00014 University of Helsinki, Finland
[3]Unit for Environmental Sciences and Management, North-West University, ZA-2520 Potchefstroom, South Africa

*Correspondence to:* A. P. Praplan (arnaud.praplan@fmi.fi)

**Abstract.** Total hydroxyl radical (OH) reactivity measurements were conducted at the second Station for Measuring Ecosystem-Atmosphere Relations (SMEAR II), a boreal forest site located in Hyytiälä, Finland, from April to July 2016. The measured values were compared with OH reactivity calculated from a combination of data from the routine trace gas measurements (station mast) as well as online and offline analysis with gas chromatography coupled to mass spectrometry (GC-MS) and offline liquid chromatography. Up to 104 compounds, mostly Volatile Organic Compounds (VOCs) and oxidised VOCs, but also inorganic compounds, were included in the analysis, even though the data availability for each compound varied with time. The averaged experimental total OH reactivity increased from April to June (from 5.3 to 11.3 s$^{-1}$) and decreased in July (8.8 s$^{-1}$) due to different environmental conditions during the measurement days. In general, the total OH reactivity increased in late-afternoon and is high at night. It decreases in the morning and is low during the day, following the pattern of mixing ratios due to change of the boundary layer height. The missing reactivity fraction (defined as the different between measured and calculated OH reactivity) was found to be high. Several reasons that can explain the missing reactivity are discussed in detail such as 1) missing measurements due to technical issues, 2) not measuring oxidation compounds of detected biogenic VOCs, 3) missing important reactive compounds or classes of compounds with the available measurements. In order to test the second hypothesis, a one-dimensional chemical transport model (SOSAA) has been used to estimate the amount of unmeasured oxidation products and their expected contribution to the reactivity for three different short periods in April, May, and July. However, only a small fraction ($< 9\%$) of the missing reactivity can be explained by modelled secondary compounds (mostly oxidised VOCs). These findings indicate that compounds measured but not included in the model as well as unmeasured primary emissions contribute the missing reactivity. In the future, non-hydrocarbon compounds from other sources than trees (e.g. soil) should be included in OH reactivity studies.

## 1 Introduction

Terrestrial vegetation is responsible for about 90 % of the emissions of Biogenic Volatile Organic Compounds (BVOCs) into the atmosphere (Guenther et al., 1995). Isoprene and monoterpenes are the most abundant BVOCs globally (44 and 17 %,





respectively; Guenther et al., 2012). These compounds are very reactive and their lifetimes range from minutes to hours, thus influencing tropospheric chemistry.

Total hydroxyl radical (OH) reactivity measurements can be used as a method to assess our understanding of tropospheric chemistry (Kovacs and Brune, 2001; Williams and Brune, 2015). Many observations of total OH reactivity have been performed in the past few decades and compared to calculated OH reactivity derived from known chemical composition of the atmosphere.

While for urban environment the unexplained (or *missing*) reactivity fraction remains low, it is often more than 50 % in forested environments (see the review by Yang et al., 2016). Based on these observations, Ferracci et al. (2018) modelled the global OH reactivity, as well as hypothetical missing chemical sink, which was found to be mostly localized above forested areas and in a few areas with large anthropogenic emissions.

Large fractions of missing reactivity were first observed in a forest in northern Michigan (Di Carlo et al., 2004) and later

observed as well in other forested environments (e.g. Hansen et al., 2014; Nakashima et al., 2014; Ramasamy et al., 2016; Zannoni et al., 2016). Also in the tropical forest of Borneo up to 70 % of the measured total OH reactivity remained unexplained (Edwards et al., 2013). In addition, Nölscher et al. (2016) identified a large difference of missing OH reactivity between the dry and wet seasons in the Amazon rainforest, with 79 % on average and between 5 to 15 %, respectively. They identified then the forest floor as an important but poorly characterized source of OH reactivity and Bourtsoukidis et al. (2018) recently identified

strong sesquiterpene emissions from soil micro-organisms at the same site.

Also in the boreal forest, which represents approximately one third of the Earth's forested surface (Keenan et al., 2015), a large discrepancy was observed between the total measured OH reactivity and the reactivity calculated from individual compounds present in the forest air (Sinha et al., 2010; Nölscher et al., 2012). Up to 89 % of the measured total OH reactivity could not be explained for periods in which the forest experienced stress conditions (elevated temperature).

The two main assumptions for the missing reactivity are 1) missing primary emissions and 2) missing oxidation products from the emissions. Several studies have been conducted to investigate these hypotheses. Nölscher et al. (2013), for instance, found an increasing missing fraction of Norway spruce (*Picea abies*) emissions from about 15–27 % in spring and early summer and up to 70–84 % in late summer and autumn. In contrast, Kim et al. (2011) found no significant unknown primary BVOC contributing to OH reactivity (for red oak, white pine, beech, and red maple) during their study period in July 2009 in

a forest in Michigan. They also found that the missing reactivity from ambient measurement at this site could be explained by oxidation products from isoprene. Kaiser et al. (2016) found in an isoprene-dominated forest in Alabama that emissions and their modelled oxidation products reduced the unexplained reactivity to 5–20 % during the day and 20–32 % at night and attribute the missing reactivity to unmeasured primary emissions. Mao et al. (2012) also demonstrated that including modelled oxidation products in OH reactivity calculations reduce the difference with measurements significantly.

Sinha et al. (2010) and Nölscher et al. (2012) conducted their studies at the second Station for Measuring Ecosystem-Atmosphere Relation (SMEAR II; Hari and Kulmala, 2005) in Hyytiälä, Finland, for about three weeks in August 2008 and for about three and a half weeks in July-August 2010, respectively, with the Comparative Reactivity Method (CRM, Sinha et al., 2008). Mogensen et al. (2011) modelled the full year of OH reactivity at SMEAR II for 2008, based on modelled emissions, known chemistry, and environmental conditions. A comparison with results from Sinha et al. (2010) showed that





compounds other than monoterpenes, isoprene, and methane contribute to only about 8 % of the measured OH reactivity. Taking all compounds into account, about 61 % of the OH reactivity remained unexplained on average during that period. Mogensen et al. (2015) also compared modelled reactivity at SMEAR II with OH reactivity measurements from Nölscher et al. (2012), using measured trace gases as input but found on average about 65 % of unexplained reactivity, similarly to the previous study.

In order to investigate OH reactivity at SMEAR II in more details, in particular its missing fraction and the seasonal variations which are often neglected for summer intensive campaigns, a new implementation of the CRM was developed at the Finnish Meteorological Institute (Praplan et al., 2017). It was installed at SMEAR II along with instrumentation to measure VOCs in spring and summer 2016.

## 2   Methods

**2.1   Measurement site**

Measurements were conducted at the boreal forest site SMEAR II (Hari and Kulmala, 2005; Ilvesniemi et al., 2009) in Hyytiälä, Finland (61°51' N, 24°17' E, 181 m above sea level). The site is located in a ca. 60-year old managed conifer forest with modest height variation of the terrain. The stand is dominated by Scots pine (*Pinus sylvestris* L.) homogeneously for about 200 m in all directions, extending to the north for about 1.2 km. Tampere is the largest city near the station about 60 km S-SW.

The instruments were located inside a container in an opening about 115 m from the site mast, from which meteorological data as well as ozone ($O_3$), nitrogen oxides ($NO_x$), methane ($CH_4$), carbon monoxide (CO) and sulfur dioxide ($SO_2$) concentrations were retrieved. In situ measurements of the total OH reactivity (section 2.5.1) and of VOC concentrations (section 2.2) were done at the container, sampling outside air at a height of about 1.5 m (Fig. 1). Station data (from the mast, measurement towers and soil) are open data under Creative Commons 4.0 Attribution licence (CC BY 4.0) and were retrieved from the online

SmartSMEAR interface (https://avaa.tdata.fi/web/smart/smear, Junninen et al., 2009).

    Temperature and relative humidity (RH) are taken at 4.2 m above ground on the mast; soil properties are an average of five locations throughout the site; and radiation and precipitation data are collected at 18 m height on a nearby tower.

**2.2   In-situ measurements of volatile organic compounds**

VOCs were measured with two in situ GC-MS. The first GC-MS was used for the measurements of mono- and sesquiterpenes,

isoprene, 2-methyl-3-butenol (MBO) and $C_{5-10}$ aldehydes. With this GC-MS air was drawn at the flow rate of $1\,l\,min^{-1}$ through a meter-long fluorinated ethylene propylene (FEP) inlet (i.d. 1/8 inch) and for $O_3$ removal (Hellén et al., 2012) through a meter-long heated (120°C) stainless steel tube (o.d. 1/8 inch). VOCs were collected from a $40\,ml\,min^{-1}$ subsample flow in the cold trap (Carbopack B/Tenax TA) of the thermal desorption unit (TurboMatrix, 650, Perkin-Elmer) connected to a gas chromatograph (Clarus 680, Perkin-Elmer) coupled to a mass spectrometer (Clarus SQ 8 T, Perkin-Elmer). A HP-5 column

(60m, i.d. 0.25 mm, film thickness 1 μm) was used for separation. The second GC-MS was used for the measurements of $C_{4-8}$ alcohols and $C_{2-7}$ volatile organic acids (VOAs). Samples were taken every other hour. The sampling time was 60 min.



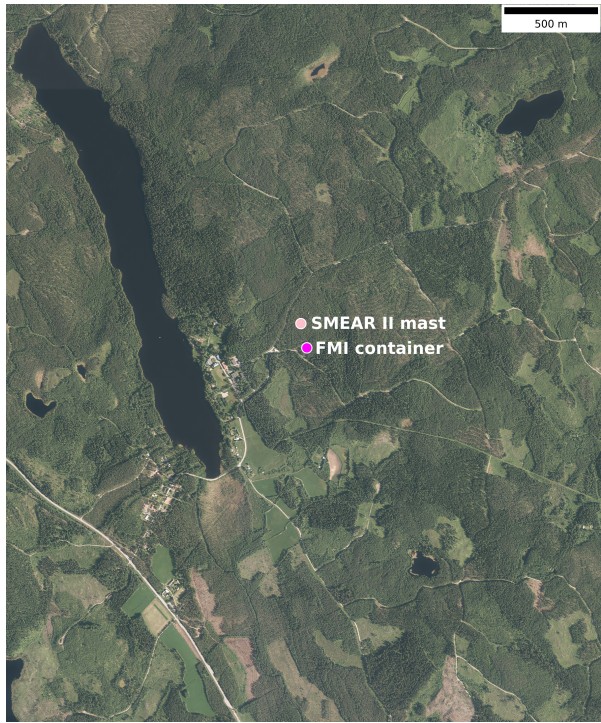

**Figure 1.** Orthophotograph of the SMEAR II station in Hyytiälä and its surroundings with the marked location of the station mast and the container where the measurements were performed. (Source: Land Survey of Finland Topographic Database 09/2018, CC BY 4.0).

Samples were analysed in situ with a thermal desorption unit (Unity 2 + Air Server 2, Markes International LTD, Llantrisant, UK) connected to a gas chromatograph (Agilent 7890A, Agilent Technologies, Santa Clara, CA, USA) and a mass spectrometer (Agilent 5975C, Agilent Technologies, Santa Clara, CA, USA). A polyethylene glycol column DB-WAXetr (30-m, i.d. 0.25 mm, a film thickness 0.25 µm) was used for the separation. These methods and measurements have been described in more detail by Hellén et al. (2017, 2018).

### 2.3 Offline measurements of volatile organic compounds

Additional sampling took place on some occasions in canisters and through adsorption cartridges (24-hour time resolution) to be analysed by GC-FID ($C_{2-6}$ hydrocarbons) and LC-UV (carbonyls), respectively. During these periods, Tenax tube samples were also taken (4-hour time resolution) and analysed later in the laboratory with GC-MS. These results were used as backup to fill in data during interruptions of the online GC-MS measurements.

### 2.4 Mixing Layer Height measurements

The Mixing Layer Height (MLH) was estimated from measurements with a 1.5 m pulsed Doppler lidar (Halo Photonics Stream Line; Pearson et al., 2009) similar to Hellén et al. (2018). MLH was determined from a combination of turbulent kinetic energy



dissipation rate profiles and conical scanning at $30\,^\circ$ elevation angle according to the method described in Vakkari et al. (2015). With this method MLH could be determined from $60\,\mathrm{m}$ above ground level (a.g.l.) to more than $2000\,\mathrm{m}$ a.g.l. at SMEAR II. Periods when MLH was $<60\,\mathrm{m}$ a.g.l. could be identified although the actual MLH was not determined due to minimum range
limitations. MLH was not determined for rainy periods. For more detailed specifications of the lidar system and the applied MLH determination method see Hellén et al. (2018).

## 2.5   OH reactivity

The OH reactivity, $R_{\mathrm{OH}}$, is defined as the sum of the concentration of individual compounds $X_i$ multiplied by their respective reaction rate coefficient with respect to OH ($k_{\mathrm{OH}+X_i}$). This can be summarised by the following equation:

$$R_{OH} = \sum_i [X_i] k_{OH+X_i} \tag{1}$$

The OH reactivity of a compound is the inverse of its lifetime with respect to OH in the atmosphere. High OH reactivity values correspond to short lifetimes and long-lived species (such as methane) have a low reactivity.

Our analysis includes up to over 100 individual species from two GC-MS, GC-FID and LC-UV measurements (see sections 2.2 and 2.3). However, not all compounds have been measured at all times (see Fig. 5c). In addition $NO_x$, $O_3$, $SO_2$ and,
and CO concentrations were retrieved from the mast of the SMEAR II station, about $115\,\mathrm{m}$ away from the sampling position of total OH reactivity and VOCs.

### 2.5.1   Total OH reactivity measurements: the Comparative Reactivity Method

Measurements of total OH reactivity ($R_{\mathrm{exp}}$) have been conducted using the Comparative Reactivity Method (CRM, Sinha et al., 2008; Michoud et al., 2015). Our particular implementation of the method is described in Praplan et al. (2017).
The CRM is based on the monitoring of pyrrole ($C_4H_5N$) mixed in a 100 ml-reactor with zero air and ambient air, alternatively. The total flow through the reactor is about $465\,\mathrm{ml\,min^{-1}}$ and the residence time in the reactor estimated about $12$–$15\,\mathrm{s}$.

Pyrrole detection is performed with a gas chromatograph (GC) equipped with a photon ionization detector (PID) every two minutes (Synthec Spectras GC955, Synspec BV, Groningen, The Netherlands). OH is produced by the photolysis of water ($H_2O$) in a nitrogen flow ($99.9999\%$ $N_2$) using ultraviolet (UV) radiation and introduced into the CRM instrument reactor.
In the zero air mixture, all OH are consumed by pyrrole ($C_2$ level), while ambient air contains other reactive compounds that compete for OH leading to a higher pyrrole concentration ($C_3$ level). The instrument switches between measurement of zero air and ambient air every 8 minutes. Stabilization of the conditions takes a couple of minutes and the first data point after each switch is discarded. From the difference between $C_2$ and $C_3$ pyrrole levels and taking into account the amount of pyrrole in the reactor in the absence of OH ($C_1$), the total OH reactivity $R_{\mathrm{eqn}}$ can be derived from the following equation:

$$R_{\mathrm{eqn}} = \frac{C_3 - C_2}{C_1 - C_3} \cdot k_p \cdot C_1 \tag{2}$$





with $k_p$ the reaction rate of pyrrole with OH ($1.2 \cdot 10^{-10}$ cm$^3$ s$^{-1}$, Atkinson et al., 1985). $C_1$ is measured by introducing a large concentration of 0.6 % propane ($C_3H_8$) in nitrogen ($N_2$) to act as an OH scavenger (Zannoni et al., 2015). Therefore, $C_1$

takes into account the photolysis of pyrrole due to the UV radiation entering the reactor (8–13 %), which decreases the pyrrole concentration from the total amount of pyrrole injected in the reactor ($C_0$ level).

Equation (2) assumes that OH levels are identical during $C_2$ and $C_3$ measurements. Therefore, variations of RH within the reactor, but also the presence of $NO_x$ and $O_3$ needs to be taken into account. Therefore $C_3$ in Eq. (2) results from the following:

$$C_3 = C_{3,\mathrm{exp}} + \Delta C_{3,\mathrm{H_2O}} + \Delta C_{3,\mathrm{NO_2}} + \Delta C_{3,\mathrm{O_3}} \tag{3}$$

with $C_{3,\mathrm{exp}}$ the measured level of pyrrole in $C_3$ mode, $\Delta C_{3,\mathrm{H_2O}}$ the correction due to different RH in $C_2$ and $C_3$ (usually small), and $\Delta C_{3,\mathrm{NO_2}}$ and $\Delta C_{3,\mathrm{O_3}}$ the corrections due to the presence in the reactor of nitrogen dioxide ($NO_2$) and $O_3$, respectively.

In addition, because of the dilution of the sampled air with humid nitrogen, the experimental total OH reactivity ($R_{\mathrm{exp}}$) is derived from the following equation:

$$R_{\mathrm{exp}} = D \cdot R_{\mathrm{CRM}} = D \cdot F \cdot R_{\mathrm{eqn}} \tag{4}$$

with $D$ the dilution factor (ratio of sampling flow over total flow through the reactor) and $F$ the correction factor for deviation from pseudo first order conditions.

Because the connection between the UV lamp and the reactor broke in June and the lamp position changed slightly after replacement of the connection, we could not use directly the corrections from Praplan et al. (2017) for the data acquired in July.

Therefore, the corrections due to the presence of $NO_2$ and $O_3$ are discussed in detail in sections 2.5.2 and 2.5.3. In addition the correction factor $F$ for deviation from pseudo first-order conditions is also discussed in detail in 2.5.4 not only due to the new lamp position, but also because of the different composition of the sampled air in this study compared to Praplan et al. (2017)

### 2.5.2 Nitrogen oxides correction factors

Praplan et al. (2017) describe the derivation of this correction in more details. Briefly, the introduction from $NO_x$ from ambient

air in the reactor causes an increase of OH in $C_3$ mode compared to $C_2$ (there $NO_x$ is removed from the catalytic converter). This is due to the photolysis of $NO_2$ to NO and the reaction of NO with $HO_2$ yielding $NO_2$ and OH.

The correction for $C_3$ ($\Delta C_{3,\mathrm{NO_2}}$) for the presence of $NO_2$ used until June (Praplan et al., 2017, from ) is plotted in Fig. 2. The uncertainty ($U_{\Delta C_{3,\mathrm{NO_2}}}$) derived from the fit is 3.7 %. For later data another correction factor was derived, due to the replacement of the UV lamp connection to the reactor after it broke, which changed the position of the lamp in the reactor's

arm. This newer correction depicted in Fig. 2 and derived at lower pyrrole-to-OH ratio (pyr:OH) is very similar to the previous correction, indicating that the new lamp position is not affecting the chemistry in the reactor much. The uncertainty ($U_{\Delta C_{3,\mathrm{NO_2}}}$) for this newer correction based on the uncertainty of the fit is 9.0 %. This higher uncertainty results from a larger variation of the signal, especially at higher $NO_2$ values that were not included in the first derivation of the correction factor.



Results from the box model described in section 2.6.1 are added for comparison. The model assume 60 % photolysis of $NO_2$ to NO and the energetically excited oxygen atom $O(^1D)$ for the data before the lamp position was changed and complete pho-

tolysis after the lamp position was changed. However, in the case with pyr:OH 1.10 the model deviates from the experimental results significantly for an unknown reason. Also note that the different corrections could be a result of a shift in pyr:OH values rather than solely be due to the different lamp position.

Both corrections take into account the change in reactivity due to the injection of $NO_2$. The correction $\Delta C_{3,NO_2}$ has been applied when it is larger than the standard deviation of $C_3$. Due to predominantly low $NO_x$, correction due to the presence of

NO was always lower than the standard deviation of $C_3$ and has therefore not been applied to the data in this study.

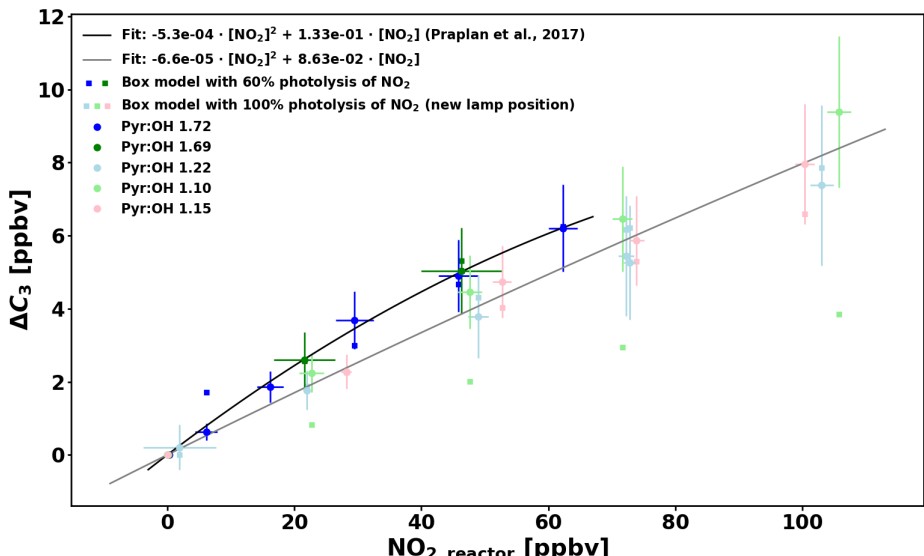

**Figure 2.** Correction of $C_3$ ($\Delta C_3$) as a function of nitrogen dioxide in the reactor ($NO_{2,reactor}$). Circles with standard deviations are experimental data and square symbols are results from the box model. The colours correspond to the same Pyr:OH as experimental data. Dark colours are from Praplan et al. (2017) and light colours are results after the UV lamp position was modified in the CRM reactor.

### 2.5.3 Ozone correction factor

As discussed in Praplan et al. (2017) and by Fuchs et al. (2017) for the CRM system of the Max Planck Institute, the pyrrole signal obtained during analysis of ambient air must be corrected for the presence of $O_3$. In the reactor $O_3$ gets photolysed producing $O(^1D)$, which reacts further with $H_2O$, yielding two OH.

Praplan et al. (2017) used a correction ($\Delta C_{3,O_3}$) independently of pyr:OH as the experimental pyr:OH for the measurements was in a narrow range close to 2. However as pyr:OH varied from 1.0 to 5.3 in this study, a pyr:OH-dependent correction has been derived.





O$_3$ correction factors ($F_{O_3}$) derived from the experimental and modelling data of Praplan et al. (2017) are depicted in Fig. 3 with dark blue and light blue markers, respectively. $F_{O_3}$ corresponds to the slope of a linear fit forced through the intercept for $\Delta C_{3,O_3}$ as a function of the O$_3$ mixing ratio in the reactor. These values are depicted in Fig. 3 as a function of pyr:OH as well as corresponding results from the box model (see section 2.6.1). Two experimental data points are labelled as outliers as discussed in Praplan et al. (2017).

Experimental data after the lamp position was altered are shown (in the lower pyr:OH range) as well as the corresponding modelling results. Good agreement could be achieved for data at pyr:OH 1.27 assuming complete photolysis of O$_3$ to O$_2$ and O($^1$D) in the reactor at about 42 % RH and 23 % photolysis at high RH (95.5 %).

A few experiments denoted by square markers in Fig. 3 were performed with propane (C$_3$H$_8$) addition in order to observe the variation of the correction at higher reactivity values.

Additionally, model experiments using ambient conditions as input were performed in order to check the correction factor value over a large spectrum of conditions. The main drawback of the model is that it assumes a degree of photolysis extrapolated from very few experimental data points. The rest of the model input is based on experimental data (from the first part of the campaign, April–June) for given pyr:OH ratios. The results are indicated with turquoise points in Fig. 3 and show some scatter as well as a plateauing trend towards high pyr:OH.

Finally, the solid black line represents a quadratic fit for all results without C$_3$H$_8$. The uncertainty on this correction ($U_{F_{O_3}}$) is 30.1 % and it takes into account variations due to the change of reactivity when acquiring data.

The correction $\Delta C_{3,O_3}$ is then derived from the following equation:

$$\Delta C_{3,O_3} = F_{O_3}[O_3] = \left(-4.92 \cdot 10^{-3}(\text{pyr}:\text{OH})^2 + 4.53 \cdot 10^{-2}(\text{pyr}:\text{OH})\right) \cdot [O_3] \tag{5}$$

It would not be feasible to derive a correction factor based on reactivity, as it is the wanted unknown quantity. Nevertheless, because the mean value for the reactivity in the reactor ($R_{\text{eqn}}$, Eq. (2.5.1)) for the campaign is about 10 s$^{-1}$, the uncertainty originating from changes in reactivity remains generally small.

### 2.5.4 First order correction factor

Sinha et al. (2008) used a two-equation model to correct for the deviation from pseudo-first-order kinetics ([Pyr]≫[OH]). Michoud et al. (2015) used more detailed modelling taking into account OH recycling reactions, but could not match the model results with their experimental data. For this reason, Michoud et al. (2015) favoured the experimental approach to correct the reactivity data. Nevertheless, the experimental approach also has drawbacks. For instance, impurities from standards and changes over time (ageing) might alter its reactivity. Also it is based on calibrations using one compound at the time, which do not represent complex ambient mixtures of reactive gases.

The reactivity calibrations were performed for the present study with a 10 ppm$_v$ C$_3$H$_8$ standard as well as with an in-house made gas mixture containing $\alpha$-pinene with small impurities from aromatic compounds. The concentrations of the C$_3$H$_8$ and





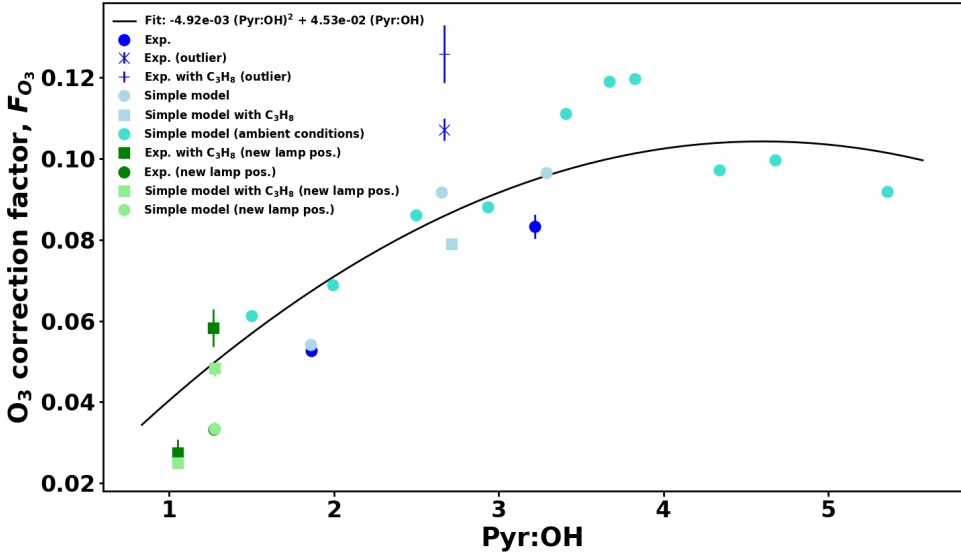

**Figure 3.** $O_3$ correction factor ($F_{O_3}$) as a function of pyr:OH for experimental data in dark colours (blue before changing the UV lamp position in the CRM reactor and green afterwards) and as model results in light colours without $C_3H_8$ (circles) and with $C_3H_8$ (squares). Turquoise circles represent model results for ambient conditions during the measurement period at SMEAR II. The fit (solid black line) includes only the results (experimental and model) without $C_3H_8$.

in-house $\alpha$-pinene standards were checked periodically by taking adsorbent tube samples and analysing them by GC-MS. At the same time impurities (4.7–17 % of the reactivity) could be measured and taken into account.

The correction factors $F$ derived from the regression slope between calculated reactivity ($R_{\text{input}}$) and measured reactivity

($R_{\text{eqn}}$) are shown in Fig. 4. Box model results (see section 2.6.1) are also included, both for the calibration conditions and for ambient conditions in order to cover a larger pyr:OH range. These ambient conditions cover both periods before and after the lamp position was changed and assume "high" and "low" input values (temperature, pressure, RH, etc.).

For $C_3H_8$, experimental $F$ values are reproduced fairly well by the model for pyr:OH values between 1.9 and 2.7, close to the parametrization (black dotted line) used by Michoud et al. (2015), who derived it from calibrations with ethane, isoprene

and propene standards. However, the model and experiment show a larger discrepancy at pyr:OH 4.3. The box model results for ambient conditions (small triangles) are slightly higher than the measured values, which are possibly due to the fact that calibration conditions include a dilution of $O_2$ and $H_2O$ due to the use of a dry $C_3H_8$ standard in $N_2$, which are not necessary in ambient conditions modelling.

For $\alpha$-pinene, some experimental values agree very well with the model, while others show a large discrepancy with the

15 model. This is particularly true for calibrations performed after the lamp position was modified. However, as the $\alpha$-pinene standard is not stable and even though concentration changes have been tracked, it is more likely that the discrepancy comes from the change in the gas mixture composition rather than different conditions in the reactor. While tracking the decreasing




concentration of $\alpha$-pinene in the standard with GC-MS, the method did not detect other compounds (such as oxidation products) and therefore the reactivity calculated for the standard is likely underestimated. Note also that no $O_2$ and $H_2O$ dilution needs to be taken into account as the in-house made gas mixture contains ambient levels of $O_2$ and $H_2O$.

5    For these reasons a fit was derived from the model results for ambient conditions reflecting an average of highly reactive compounds (such as monoterpenes) and less reactive compounds and has been used to correct ambient data. The uncertainty of this correction ($U_F$) derived from the uncertainty of the fit is 10.0 %.

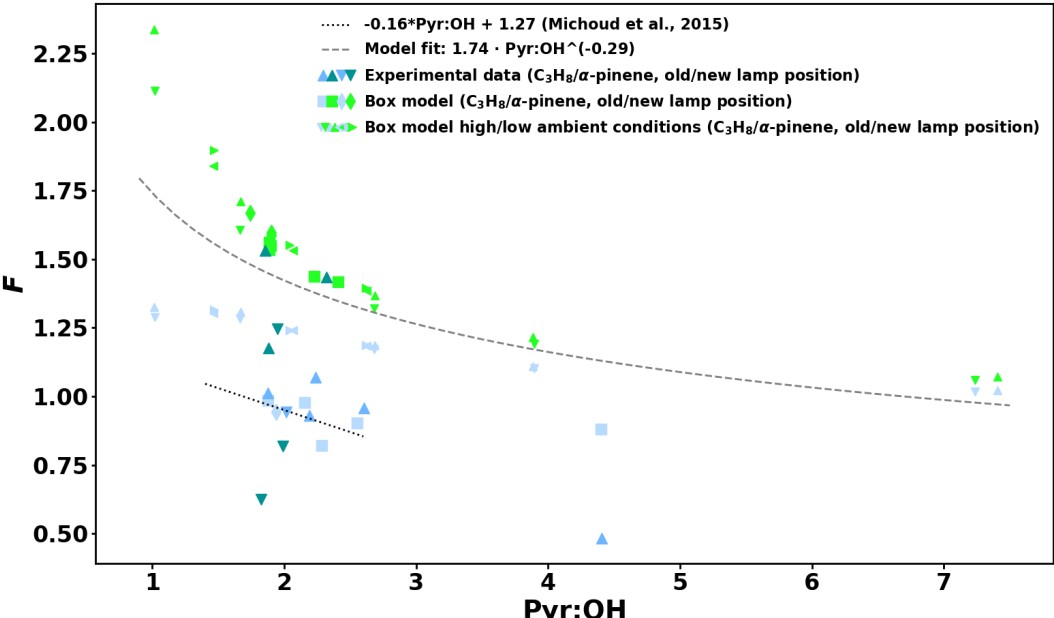

**Figure 4.** Pseudo-first-order correction factor, $F$, as a function of pyr:OH. Experimental data are represented with dark coloured large triangles in blue for $C_3H_8$ data and in green for $\alpha$-pinene and pointing up for the original lamp position and pointing down after the lamp position was changed. Box model results for the calibration are indicated in light colours with square for the original lamp position and with lozenges for the later lamp position. Model results for ambient conditions are represented with smaller symbols and result in the dashed-line fit. For comparison, the correction from Michoud et al. (2015) is indicated with a dotted line.

The calibration factor $F$ is derived from the calculated pyr:OH and then used to derive $R_{\mathrm{CRM}}$ according to the following equation:

$$R_{\mathrm{CRM}} = F \cdot R_{\mathrm{eqn}} = 1.74 \cdot (\mathrm{pyr:OH})^{-0.29} \cdot R_{\mathrm{eqn}} \tag{6}$$

This parametrization leads to larger corrections for measurements at lower pyr:OH. Also it has been shown that this correction depends on the reactivity of the calibration gas used (Michoud et al., 2015). Therefore neither approach (experimental or





model) is able to fully capture the complexity of the chemistry in the CRM reactor under ambient conditions, where a large
variety of compounds react with OH and other oxidants. Nevertheless, results neglecting this correction factor ($R = D \cdot R_{\mathrm{eqn}}$)
are plotted alongside results using this correction to illustrate how it affects the results (see section 3.1 for a more detailed
discussion).

## 2.6    Models

### 2.6.1    Box model for the CRM reactor

The chemistry in the CRM instrument's reactor was simulated by a box model. It is based on the inorganic section of the
Master Chemical Mechanism (MCM, http://mcm.leeds.ac.uk/MCM/) in its version 3.3.1 with amendments by Michoud et al.
(2015). Minor improvements, such as varying temperature, pressure and RH have been implemented in addition. Also, instead
of scaling the OH concentration in order to match the modelled $C_2$ with the experimental value, $O(^1D)$ is added (from the
photolysis of the formed $HO_2$) to the initial conditions to make both $C_2$ values match. This approach leads to an increase in
$O_3$ (higher values at lower pyr:OH), which has been measured in the CRM reactor previously (Sinha et al., 2008; Michoud
et al., 2015) as well as in our system (about 170 ppbv). Also instead of considering only the reactions of the VOCs of interest
with OH (e.g. $\alpha$-pinene and $C_3H_8$ for the derivation of $F$) as done by (Michoud et al., 2015), it includes only first generation
of reactions extracted from MCM for these compounds because of the short residence time in the CRM reactor. This becomes
relevant especially for unsaturated compounds (such as $\alpha$-pinene) due to the presence of $O_3$ in the reactor.

This box model is far from taking into account the complex processes in the CRM reactor, but it is a useful tool to test
hypotheses (such as $NO_2$ and $O_3$ photolysis) and to extend the validity range of correction factors that depend on pyr:OH to
conditions that were not available experimentally.

### 2.6.2    SOSAA

In this study we applied the model to Simulate the concentrations of Organic vapours, Sulphuric Acid and Aerosols (SOSAA) to
simulate the OH reactivity at the SMEAR II station for selected days in April, May, and July 2016. SOSAA is a one-dimensional
chemical transport model comprised of boundary layer meteorology, biogenic emission of VOCs, gas-phase chemistry, aerosol
dynamics and gas dry deposition (e.g. Boy et al., 2011; Zhou et al., 2014) and has been previously used to simulate OH
reactivity at this site (Mogensen et al., 2011, 2015).
The boundary layer meteorology was derived from SCAlar DIStribution (SCADIS; Sogachev et al., 2002), as described in
Boy et al. (2011). The biogenic emission module was deactivated because in situ measurements were used to provide input
concentrations. Biogenic compounds were set to the measured values up to 18 m (canopy height), while aromatic compounds
were set to the measured values at all heights. Measured inorganic gas concentrations at SMEAR II were used as input. The
gas-phase chemistry was created using the Kinetic PreProcessor (KPP; Damian et al., 2002). The chemical reaction equations
used in this study were selected from the Master Chemical Mechanism v3.3.1 (MCMv3.3.1 Jenkin et al., 1997; Saunders et al.,
2003; Bloss et al., 2005; Jenkin et al., 2012, 2015). The chemistry scheme included more than 15000 reactions, and a total of



3525 chemical species representing the complete reaction paths for isoprene, $\alpha$-pinene, $\beta$-pinene, limonene, $\beta$-caryophyllene,

methane, 2-methyl-3-buten-2-ol (MBO), benzene, toluene, styrene, ethylbenzene, 1,2-dimethylbenzene, 1,3-dimethylbenzene, 1,4-dimethylbenzene, 1,2,3-trimethylbenzene, 1,2,4-trimethylbenzene, 1,3,5-trimethylbenzene, 1-ethyl-2-methylbenzene, 1-ethyl-3-methylbenzene, 1-ethyl-4-methylbenzene, heptane, octane, nonane, butanal, pentanal, methacrolein and relevant inorganic reactions. First order reactions between OH, $O_3$, and $NO_3$ with the following monoterpenes were also included in the chemistry: $\Delta^3$-carene, myrcene, camphene and 1,8-cineole. Likewise, first order reactions between OH, $O_3$, $NO_3$ and $\beta$-

farnesene were included. The photochemistry has been improved by calculating the photodissociation constants more precisely using data from Atkinson et al. (1992). The OH reactivity has been calculated similarly as in Mogensen et al. (2011, 2015). The condensation sinks for sulfuric acid and nitric acid, based on Differential Mobility Particle Sizer (DMPS) and Aerodynamic Particle Sizer (APS) data from SMEAR II, were included (Boy et al., 2003). Since sulfuric acid and nitric acid make up most of the condensation sinks, sinks of VOCs into the particle phase are not taken into account, thereby the aerosol module is turned

off.

    The model runs in the present study include the dry deposition module implemented in SOSAA by Zhou et al. (2017a) and extended in Zhou et al. (2017b). The latter describe the explicit simulation of the loss of every compound in the model by dry deposition inside the canopy for all height levels.

## 3 Results and discussion

### 3.1 Overview

An overview of the measured total OH reactivity together with the calculated OH reactivity from up to 104 compounds, depending on data availability, as well as selected ancillary data, such as environmental conditions (air and surface soil temperatures as well as surface soil water content), and contributions from different compounds and groups of compounds are presented in Fig. 5. The following sections are discussing in details various aspects of the results such as a) seasonality, b) diurnal variations,

and c) missing reactivity. Nevertheless, from this overview, the following observations can be made:

- The range of measured total OH reactivity values is similar to previous studies at the same site in August 2008 and July-August 2010 (Sinha et al., 2010; Nölscher et al., 2012).

- The calculated OH reactivity from measured compounds is in general lower than the measured total OH reactivity (also for periods with a large number of compounds included in the analysis), leading to a large fraction of *missing* reactivity.

- A few total OH reactivity peaks in the spring (with values higher than at the end of July) seems to be associated with changes in the soil water content.

- Inorganic compounds ($CH_4$, CO, $O_3$, and $NO_2$) form an important fraction of the calculated OH reactivity.



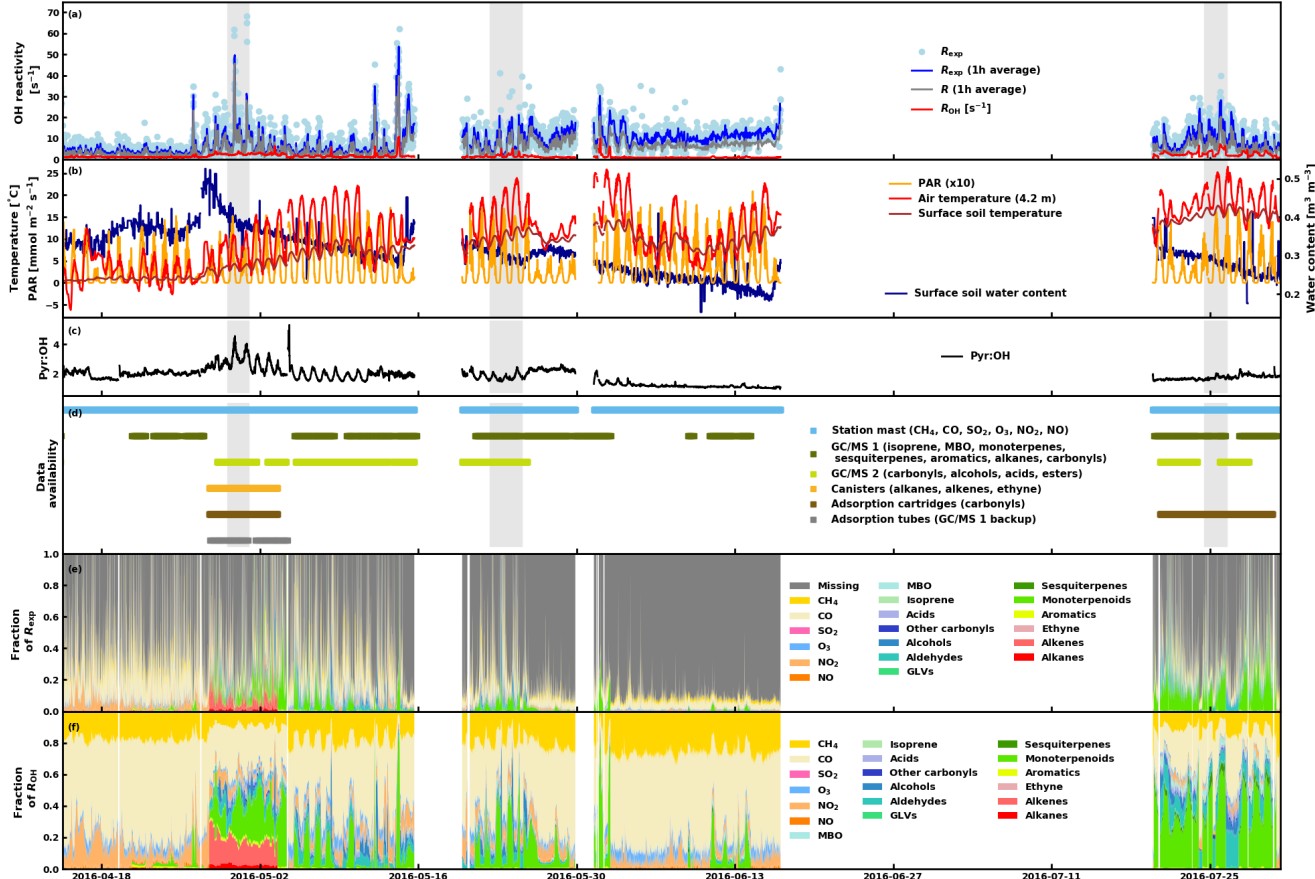

**Figure 5.** a) Experimental total OH reactivity $R_{\mathrm{exp}}$ (and its 1-h average), 1-h average of experimental total OH reactivity without pseudo-first-order kinetic correction, $R$, and calculated OH reactivity $R_{\mathrm{OH}}$, b) environmental conditions (air and surface soil temperatures, as well as surface soil water content), c) Pyr:OH in the CRM reactor, d) data availability from the different instrumentation/sources, e) fraction of experimental total OH reactivity, and f) fraction of calculated OH reactivity. The periods shaded in gray in panels (a) to (d) represent the periods investigated with SOSAA (see sect 3.4).

## 3.2 Total OH reactivity

5  Keeping in mind that the experimental data have not always been acquired continuously, the total experimental OH reactivity ($R_{\mathrm{exp}}$) monthly mean increased from April (5.4 s$^{-1}$ for 17 days) to June (11.3 s$^{-1}$ for 16 days) and decreased slightly again in July (9.0 s$^{-1}$ for 12 days) when the mean RH for the measurement period in that month increased to values similar to the measurements in April and the photosynthetically active radiation (PAR) decreased (Table 1). The data for July cover days that were cloudier and more humid (both air and soil) than the period covered by the data in June. Monthly means of ambient concentrations of locally emitted terpenoids had exponential correlation with temperature (see also Hellén et al., 2018) and a similar weak correlation exists between $T$ and $R_{\mathrm{exp}}$ (the exponential regression $y = a \cdot \mathrm{e}^{bx}$ with $a = 5.2$ and $b = 0.039$



**Table 1.** Monthly means and standard deviations (std.) of experimental total OH reactivity ($R_{\text{exp}}$), the missing OH reactivity fraction ($R_{\text{missing}}$), monoterpene and sesquiterpene mixing ratios ([MT] and [SQT], respectively), Photosynthetically Active Radiation (PAR), precipitation (Precip), relative humidity (RH), air temperature ($T$), surface soil temperature ($T_{\text{soil,humus}}$), surface soil water content ($w_{\text{soil,humus}}$), and Mixing Layer Height (MLH). Coefficients $a$ and $b$ from linear regressions between the weekly means of these variables and weekly averaged $R_{\text{exp}}$ and the corresponding coefficients of determination ($r^2$). $n_{\text{days}}$ indicates the number of days with measurements. $n$ denotes the amount of $R_{\text{exp}}$ observations. Note that all other means (except MLH) have been derived for the same measurement period as $R_{\text{exp}}$. $n_{\text{MLH}}$ indicates the amount of observations with overlapping $R_{\text{exp}}$ and MLH measurements.

| | April mean (std.) | May mean (std.) | June mean (std.) | July mean (std.) | Linear regressions ($ax+b$) $a$ | $b$ | $r^2$ |
|---|---|---|---|---|---|---|---|
| $n_{\text{days}}$ | 17 | 26 | 16 | 12 | | | |
| $n$ | 1452 | 2201 | 1421 | 973 | | | |
| $R_{\text{exp}}$ (s$^{-1}$) | 5.3 (6.4) | 7.9 (6.9) | 11.3 (4.2) | 8.8 (5.7) | | | |
| $R_{\text{missing,fraction}}$ | 0.56 (0.26) | 0.64 (0.29) | 0.90 (0.07) | 0.59 (0.26) | 0.04 | 0.33 | 0.69 |
| [MT] (ppt$_v$) | 72.4 (163.1) | 205.3 (460.1) | 83.3 (407.5) | 559.1 (505.5) | 9.6 | 117.3 | 0.02 |
| [SQT] (ppt$_v$) | 0.071 (0.275) | 1.86 (2.77) | 1.12 (3.77) | 22.9 (23.6) | 0.36 | 1.81 | 0.02 |
| PAR (µmol m$^{-2}$ s$^{-1}$) | 245.7 (332.3) | 414.6 (478.1) | 491.3 (521.1) | 362.2 (423.1) | 19.0 | 226.1 | 0.27 |
| Precip (mm) | 0.12 (0.09) | 0.12 (0.13) | 0.12 (0.18) | 0.10 (0.01) | 0.004 | 0.088 | 0.17 |
| RH (%) | 79.2 (20.3) | 62.3 (24.4) | 57.9 (21.1) | 78.9 (16.0) | -0.5 | 71.7 | 0.01 |
| $T$ (°C) | 3.6 (3.6) | 12.7 (4.8) | 12.3 (5.6) | 18.0 (3.5) | 1.1 | 2.4 | 0.37 |
| $T_{soil,humus}$ (°C) | 1.4 (1.1) | 8.1 (2.3) | 9.9 (2.1) | 15.2 (1.4) | 1.1 | -1.0 | 0.45 |
| $w_{soil,humus}$ (m$^3$ m$^{-3}$) | 0.37 (0.04) | 0.32 (0.03) | 0.24 (0.03) | 0.28 (0.03) | -0.01 | 0.41 | 0.49 |
| $n_{\text{MLH}}$ | 1431 | 2180 | 1295 | 966 | | | |
| MLH (m) | 287.1 (386.2) | 499.7 (694.9) | 574.2 (679.3) | 314.3 (447.0) | 15.8 | 303.5 | 0.09 |

has a higher coefficient of determination $R^2$, 0.56, as the linear regression), indicating that temperature-dependent biogenic

5   emissions are an important driver of the total measured OH reactivity, which has also been observed earlier (e.g. Nakashima et al., 2014; Ramasamy et al., 2016)

     The strongest correlation for weekly means was found between $R_{\text{missing,fraction}}$ and $R_{\text{exp}}$, indicating that occurrences of high reactivity correlates with higher missing reactivity. A (negative) correlation was found for weekly means between $R_{\text{exp}}$ and the surface soil water content ($w_{\text{soil,humus}}$) and also a (positive) correlation was found between $R_{\text{exp}}$ and soil surface temperature

10   ($T_{\text{soil,humus}}$). Therefore higher soil water content values corresponded to lower total OH reactivity values, indicating that a wet (and cold) soil act as sinks for reactive compounds in line with findings from Nölscher et al. (2016). However, high reactivity values were observed in spring even with low temperatures and low emissions from local vegetation (see Fig. 5b-c). This happened when the surface soil water content was the highest as the surface soil temperature started to increase above 1.5 °C, indicating thawing of the soil, a possible source of OH reactive compounds. Forest floor emissions of monoterpenes are known to be high in spring after snow has melted (Hellén et al., 2006; Aaltonen et al., 2011; Mäki et al., 2017) and VOC emission



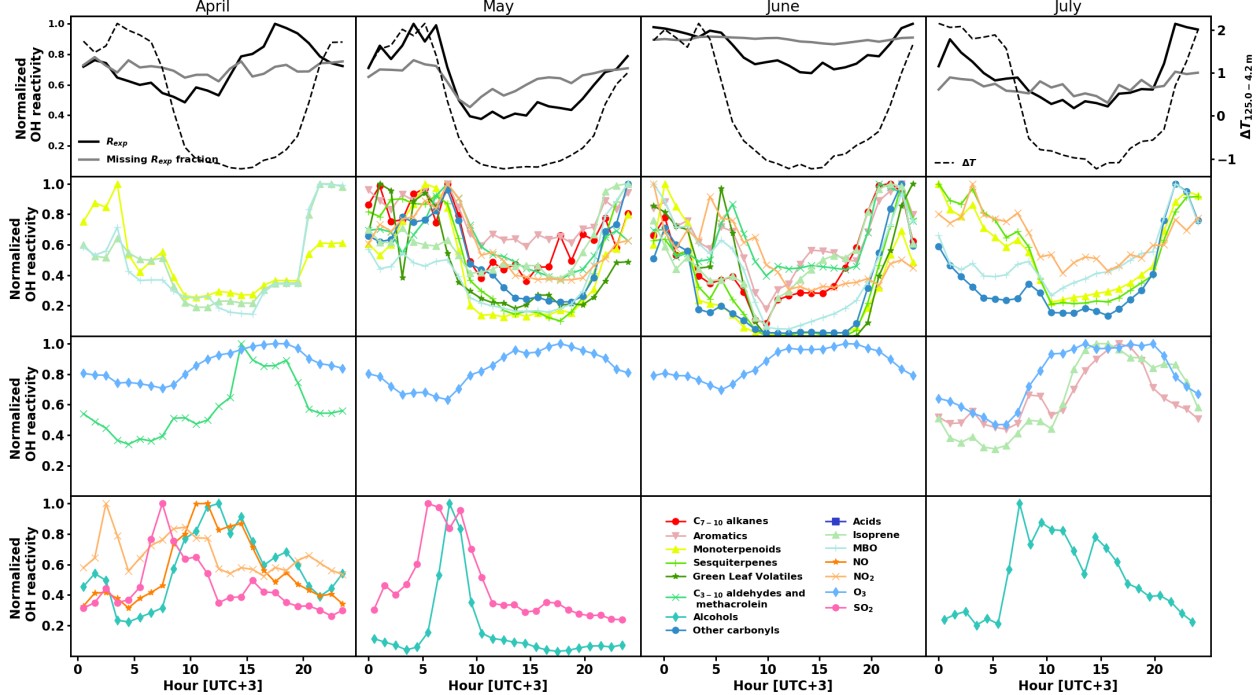

**Figure 6.** Normalized monthly averaged diurnal variations of experimental OH reactivity $R_{\mathrm{exp}}$ and the missing fraction as well as temperature gradient between 4.2 and 125.0 m above ground as a proxy for mixing layer height (top row), and calculated OH reactivity separated by group of compounds (second to fourth row).

bursts have been observed after wetting events (e.g. Rossabi et al., 2018). There has also been some indication that thawing snow/soil could be a source of volatile organic amines (Hemmilä et al., 2018). In the present study, the soil was snow-free already on 8 April, but a short snowfall episode happened later with 5 cm of snow measured on the morning of 25 April (which were gone on the next day). This episode happens just before the first OH reaction peak (at about $30\,\mathrm{s^{-1}}$), but this single occurrence is too little information to conclude of the role of snow in the large OH reactivity values observed and it might well be due to a combination of factors (including snowfall and immediate melting).

### 3.3   Diurnal variations

The calculated OH reactivity of various groups of compounds shows different diurnal patterns, which vary with the season as well. Their normalized values are depicted in Fig. 6 (second to fourth row), separated by month (April to July in columns), together with the normalized diurnal patterns of $R_{\mathrm{exp}}$ and its missing fraction and temperature difference between measurements at 4.2 m and 125.0 m above ground as a proxy for mixing layer height (top row). Compounds that had a 24-hour sampling time were removed from this analysis. Sinha et al. (2010) did not measure a clear OH reactivity diurnal pattern during their two-week measurement period and the modelling of the OH reactivity also showed no diurnal pattern (Mogensen et al., 2011).





However, Mogensen et al. (2015) modelled a weak diurnal pattern with a maximum at night, mostly due to improvements in the meteorological scheme. The observations in the present study, even though at higher OH reactivity levels show this pattern from May to July. Nölscher et al. (2012), for measurements during the same period, however, identified a similar diurnal pattern with maximum at night during the identified stress period. For normal boreal forest conditions, they measured large variations in the afternoon reactivity, sometimes leading to a maximum, sometimes not.

When the total measured OH reactivity hourly average is at a minimum during the day and a maximum at night (May to July), it follows the pattern of BVOCs concentrations (and calculated OH reactivity) due to the low mixing layer height and despite slightly lower emissions due to the lower temperatures at night (Hellén et al., 2018). In April the hourly average of missing reactivity fraction oscillated between 62.7 and 79.4 % and in June between 84.6 and 92.3 %, without a clear diurnal pattern. This is due to the fact that during these periods, only few compounds could be included in the calculated OH reactivity value and that the total OH reactivity values measured in June were higher than in April. On the other hand, for the months of May and July (when more compounds could be included in the calculated OH reactivity values) the missing fraction was lower during the day and higher at night. This possibly indicates that the oxidation products formed at night accumulate due the very low OH levels. The missing fraction of the hourly averages varied between 39.5 and 77.4 % in May and between 45.1 and 72.5 % in July, similar to values from Nölscher et al. (2012) and despite the inclusion of more compounds in our analysis.

While the OH reactivity daily patterns from monoterpenoids and MBO had a minimum during the day for all months, other groups of compounds showed this reactivity pattern only for some periods. Isoprene showed this pattern except in July, where the light-induced emissions during the day were dominating. Sesquiterpenes, other carbonyls and $NO_2$ showed a similar pattern with daytime minima from May to July, while $C_{7-10}$ alkenes, aromatics, $C_{3-10}$ aldehydes, and methacrolein showed a pattern with daytime minimum only in May and June.

Alcohols exhibit an OH reactivity pattern with a maximum in the morning (9-11 a.m.). The absolute OH reactivity of alcohols is low and dominated by 1-butanol, which is used in aerosol measuring devices at the site. It is not clear what causes the diurnal pattern, but $SO_2$ reactivity had a similar pattern in April and May, and $NO_x$ had such a pattern in April, when the photochemistry is not yet very strong.

Overall, from May to July the total OH reactivity exhibits a minimum during the day and a maximum at night, following the OH reactivity pattern for biogenic compounds (except for isoprene in July, which is present in low concentrations in this pine forest, and has a maximum in the afternoon then). In April, the total OH reactivity has a maximum in the afternoon, but no measured group of compounds display a similar diurnal pattern pointing towards unknown primary emissions of non-terpene compounds (e.g. from soil).

## 3.4 Missing OH reactivity

The comparison between the calculated and measured OH reactivity is challenging as the calculated values are derived from a number of compounds that varies because of the availability of the measurements (Fig. 5d). Some periods include only a few inorganic compounds from the station mast while other periods include a large amount of (O)VOCs analysed by the GC-MSs. The contribution to the known reactivity is shown in Fig. 5f. Even with the maximum number of compounds used to calculate





OH reactivity a large fraction of the measured total OH reactivity remain unexplained (*missing* reactivity, Fig. 5e). This fraction is similar to previous observations at this site (Sinha et al., 2010; Nölscher et al., 2012) despite adding more compounds such as sesquiterpenes to the analysis.

During a week in late April/early May additional compounds were sampled with offline methods and subsequently analysed in the laboratory as described earlier. This period coincided with high reactivity peaks observed likely due to soil thawing as mentioned previously. No compound or group of compounds that was measured during this period was peaking at the same time as the total OH reactivity.

Three scenarios can be presented from our dataset regarding missing OH reactivity:

1. Only few compounds are included in the analysis, leading to a high missing fraction (0.76 on average taking into account the beginning of the measurements period and measurements after 25 May until mid-June). This fraction would be reduced by including additional compounds in the calculated values (measured or modelled).

2. Many compounds are included in the analysis, but their oxidation products are not measured directly, which is likely the case in May (3 to 25) and July with an average missing fraction of 0.60. This fraction can be decreased by including
modelled oxidation compounds using measured mixing ratios as input in some cases (see below).

3. Many compounds are included in the analysis (including modelled compounds), but the compounds included do not represent the right class of compound(s) responsible for the OH reactivity. This is most likely the case for the intense measurement period between 27 April and 3 May with an average missing fraction for the reactivity of 0.52.

To test the hypotheses in these scenarios, three periods of two to three days for the months of April, May and July were
simulated with the SOSAA model using measured trace gases and meteorological conditions as inputs (see Section 2.6.2). The results for the inclusion of modelled oxidation compounds in the analysis are presented in Fig. 7. These compounds labelled modelled OVOCs are mostly peroxides, alcohols, and carbonyl compounds due to the generally low $NO_x$ levels at the site. Modelled inorganics, whose contributions is negligible, regroup molecular hydrogen ($H_2$), hydrogen peroxide ($H_2O_2$), nitrous acid (HONO), peroxynitric acid ($HO_2NO_2$), nitric acid ($HNO_3$), and the nitrate radical ($NO_3$).
While the trend of $R_{OH,model}$ follow the general trend of $R_{exp}$, $R_{OH,model}$ usually underestimate $R_{exp}$, especially at night. Total OH reactivity values are in general lower during the day and they are closer to $R_{OH,model}$ values, considering the scatter of the experimental data. In April, the high peaks in the late afternoon of 29 and 30 April indicate missing primary emissions, which also contribute (or their oxidation products) to the missing reactivity in the following nights.

Retrieving the additional reactivity from these modelled compounds that were not included in $R_{OH}$ reduced the missing reactivity by only a small fraction (about 8.4 % for the studied period in July and less for the other periods) as seen in Fig. 8. A detailed breakdown of the individual compounds contributing to the reactivity and their mixing ratios can be found in the Appendix.

Most of the missing reactivity could be then due to oxidation products that are not included in the model from measured
precursors such as $\Delta^3$-carene, myrcene, camphene, 1,8-cineol, $\beta$-farnesene, or unidentified sesquiterpenes, but the contribution





to the OH reactivity from these precursors is small due to their low atmospheric concentrations, so that the contribution from their oxidation products is also expected to be small (Hellén et al., 2018). The remaining missing reactivity could be also explained by oxidation products that were deposited and re-emitted from surfaces (so that they would not be taken into account when modelling their concentrations from atmospheric production based on their precursors concentrations). As mentioned earlier, missing primary emissions also contribute to the missing reactivity, more so in spring than in summer.

A previous study by Mogensen et al. (2011) modelled the OH reactivity at the SMEAR II site for the year 2008, using modelled emissions, and estimated the OH reactivity to be about $2\text{--}3\,\mathrm{s}^{-1}$ between April and July. This is lower than the measured averages from the present and previous studies and lower than the night-time modelled values in July from the present study. Mogensen et al. (2011) report that secondary organics, $\beta$-caryophyllene, farnesene, and MBO represent 8 % of the total OH reactivity, which represent the same magnitude as the results from this study. Mogensen et al. (2015) modelled the OH reactivity at the same site for July and August 2010 with the same methodology (including minor model improvements) and obtained values between 2.7 and $3.2\,\mathrm{s}^{-1}$. The higher modelled values in our study indicates that modelled emissions lead to lower monoterpene concentrations than measured concentrations.

Our results are not entirely in line with other studies that showed reductions of the missing reactivity by constraining VOC concentrations to model their oxidation products (e.g. Mao et al., 2012; Edwards et al., 2013; Kaiser et al., 2016), as the reduction observed remains small in this study. This approach still leaves a large unexplained fraction of OH reactivity. This is a strong indication that on one hand non-terpenoid compounds or re-emitted oxidation products contribute to the total OH reactivity and that on the other hand more compounds have to be included in the chemical model.

Finally, heterogeneous loss of OH to particles might be a contribution to missing OH reactivity, but this process is poorly quantified (Donahue et al., 2012). Due to the low sampling flow and long FEP sampling line to the CRM instrument, it is unlikely that particles will reach the reactor. Additionally, we could not find any correlation between ambient particle numbers and either total measured OH reactivity or its missing fraction.

As a side note, total OH reactivity measurements were unfortunately not available in the autumn, but Liebmann et al. (2018) who measured nitrate radical ($NO_3$) reactivity at the same site made similar findings, with about 30 % of unexplained $NO_3$ reactivity at night and about 60 % during daytime. Mogensen et al. (2015) modelled $NO_3$ reactivity at the site and found a maximum in the early morning, while the measurements from Liebmann et al. (2018) showed a maximum at night. The modelled $NO_3$ reactivity values were similar to the measured ones without strong temperature inversion at night, while higher measured values were recorded for nights with strong temperature inversion.

Hellén et al. (2018) showed that the balance between the emissions of VOCs and the production of oxidation compounds and the sinks vary with the season, leading to different diurnal profiles for compounds such as isoprene, $C_{7-10}$ aldehydes, and nopinone. This can also be observed in terms of OH reactivity in the present study (see section 3.3).

### 3.5 Inhomogeneity of forest air composition

As observed in Praplan et al. (2017), inhomogeneity of the air composition at the sampling site can affect the comparison between experimental total OH reactivity and calculated reactivity from known composition. It can for instance be directly



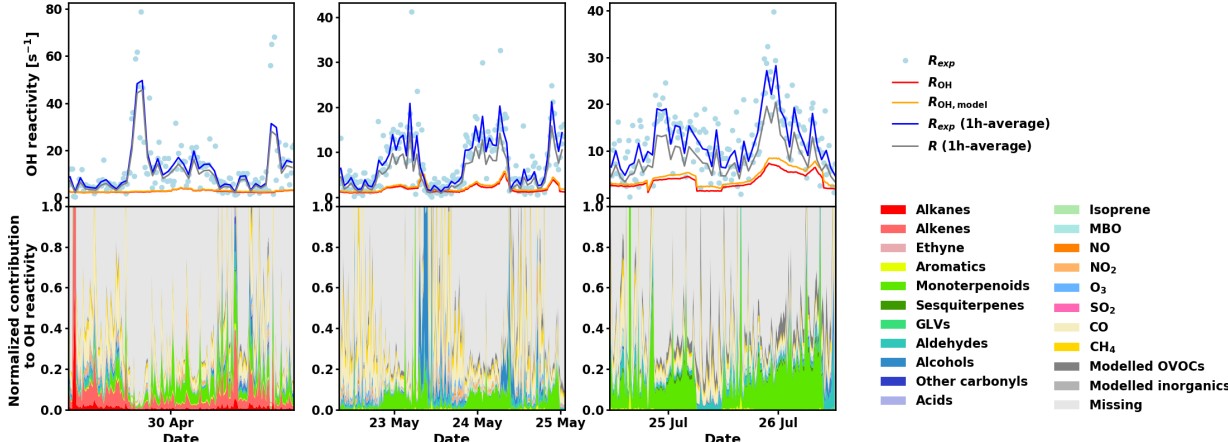

**Figure 7.** Measured total OH reactivity ($R_{\mathrm{exp}}$), calculated OH reactivity from measured compounds ($R_{\mathrm{OH}}$), calculated OH reactivity including measured and modelled compounds ($R_{\mathrm{OH,model}}$) and 1 hour averages of $R_{\mathrm{exp}}$ and $R$, the measured total OH reactivity without pseudo-first-order kinetics correction, (top panels) and normalized contributions to $R_{\mathrm{exp}}$ for various compounds and group of compounds (bottom panels) for the three periods investigated with SOSAA (see main text for details). The group labelled "Model" refers to compounds that were not directly measured, but modelled from their precursor concentrations and environmental conditions and the values larger than 1 (when $R_{\mathrm{OH,model}} > R_{\mathrm{exp}}$) have been cropped for clarity.

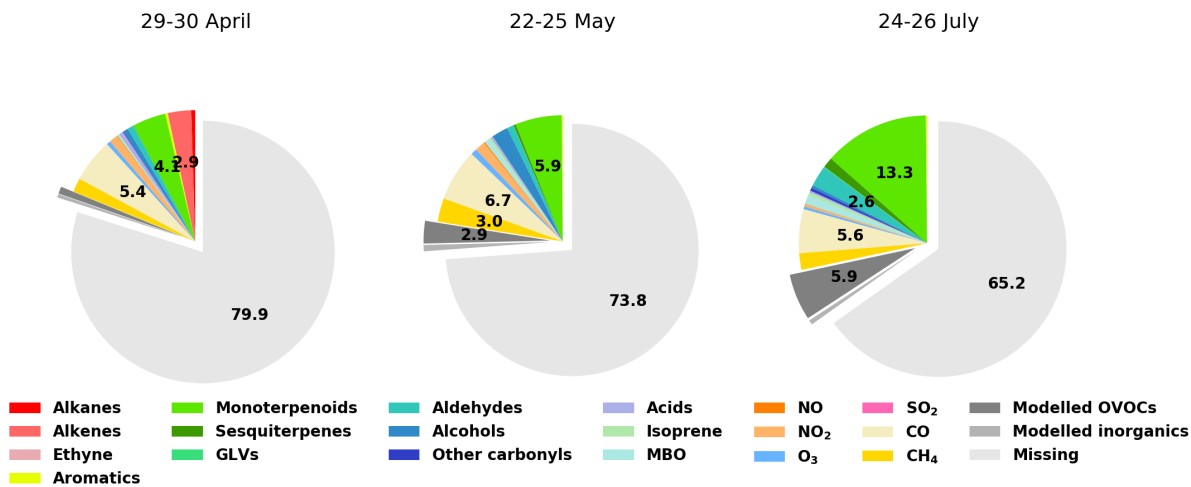

**Figure 8.** Contributions of various compounds and groups of compounds to the measured total OH reactivity ($R_{\mathrm{exp}}$). The group labelled "Model" refers to compounds that were not directly measured, but modelled from their precursor concentrations and environmental conditions. For clarity, labels for fractions smaller than 2.0 % have been omitted.



affected by meteorology or changes in concentrations between the various sampling locations due to local emissions during low mixing periods (see  Liebmann et al., 2018). As VOCs in this study were sampled at the same location than the total OH reactivity, the effect of inhomogeneity of the air composition is minimized. However, the ozone mixing ratio used to derive the ozone correction (described in section 2.5.3) is retrieved from the station mast (115 m away) and at a height of 4.2 m. It is very likely that emissions from soil and understorey vegetation (or from standing water close to the OH reactivity sampling

location) would further deplete the ozone close to the ground, leading to an overestimation of the correction.

On 29 and 30 April total OH reactivity peaks exceeding $60\,\mathrm{s^{-1}}$ in the afternoon are followed by $O_3$ concentration drops below canopy (Fig. 9) as described in Chen et al. (2018). While the high reactivity peaks themselves are likely not affected by an overestimation of the correction, the period following them (night-time) might be slightly overestimated due to the sampling of $O_3$ further away and higher above ground. This effect is difficult to take into account in retrospect. The concentration of $O_3$

should have been measured immediately next to the CRM system. Similar conditions were observed during nights between 11 and 16 May and to some extent in July (without reaching such high total OH reactivity values as in spring). This effect on the inhomogeneity of the forest air composition might affect total OH reactivity measurements and in turn partly explain some of the missing fraction.

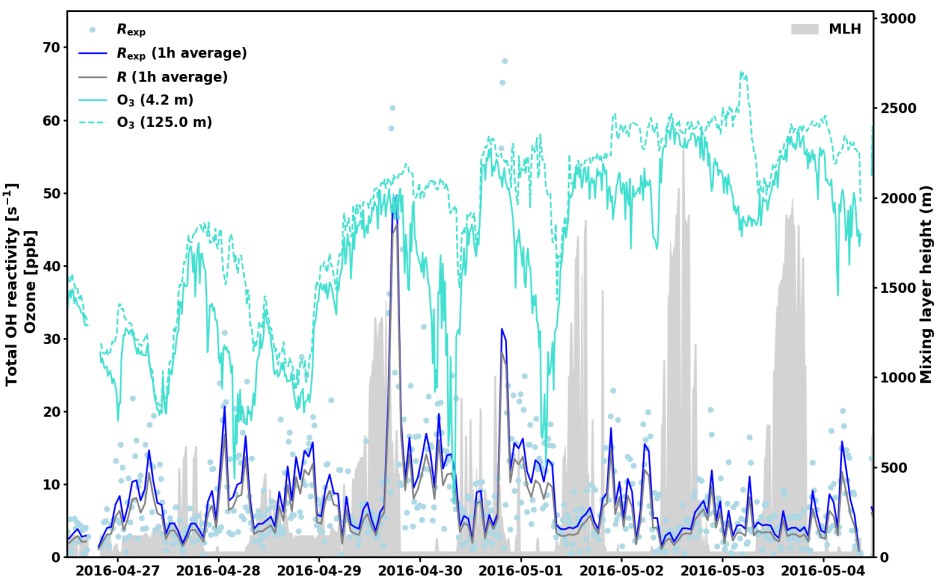

**Figure 9.** Total measured OH reactivity, $R_{\mathrm{exp}}$, and its 1h-average, as well as $R$ (1h-average), and ozone mixing ratios at 4.2 and 125.0 m above ground. Mixing Layer Height (MLH) is shown as a gray shadow. Note that the detection limit for MLH is 60 m and values below this limit are displayed at 30 m (and zeros denote gaps in the data).





## 4 Conclusions

Total OH reactivity is not a simple function of a few variables. It includes many complex processes involving sources and sinks that can change dramatically depending on the environmental conditions and the time of the year. Data availability for comparison between measured total OH reactivity values and calculated values also represent a challenge when interpreting results.

In the present study total OH reactivity measurements were performed at a Finnish boreal forest research site (SMEAR II). The averaged experimental total OH reactivity increased from April to June before decreasing in July because of more humid nights and lower radiation during the measurement period. The total OH reactivity diurnal pattern from May to July follows the one of biogenic compounds with high values during the night due to the low mixing height, even though emissions are lower at night.

A suite of online and offline (O)VOCs measurements was used to calculate the known fraction of OH reactivity to compare it to the total OH reactivity measured. The missing fraction of the OH reactivity was also higher during the night, possibly due to a larger fraction of non-measured oxidation products, compared to day time, when the emissions are higher resulting in a larger fraction of known precursors. Oxidation products resulting from $O_3$ oxidation at night are not lost chemically (due to the very low levels of OH), which might explain the higher missing fraction of OH reactivity observed at night.

Nevertheless, as the data availability of (O)VOCs varies, the comparison between experimental and calculated OH reactivity is difficult but three different explanations can lead to high missing (unexplained) OH reactivity: 1) simply the lack of measurements, 2) not measuring oxidation products (only their precursors), and 3) not measuring the right class of compounds. Using one-dimensional transport model to estimate oxidation products concentrations from measured precursor concentrations for three short periods of two to three days in various months (with most (O)VOC data availability) it is demonstrated that only a small fraction (up to ca. 9 %) of the missing reactivity can be explained by these oxidation products. On one hand, this is due to the absence in the model of degradation scheme for detected compounds in the ambient air (e.g. $\Delta^3$-carene, $\beta$-farnesene), but on the other hand it is also possible that non-hydrocarbon compounds contribute to the OH reactivity as well. However, it might not be completely excluded that re-emissions of oxidation products of terpenes from surfaces are causing increases in OH reactivity. The model does not take into account this effect, as it only estimates concentrations of oxidation products based on the concentrations of their precursors.

More measurements of oxidised compounds and identification of non-terpene reactive compounds from emissions also from other sources than vegetation (e.g. soil) are required to better understand the reactivity and local atmospheric chemistry in the forest air in general, in particular during winter, spring, and autumn, when the forest air chemistry is not dominated by emissions from the vegetation.

*Author contributions.*

A. P. Praplan conducted total OH reactivity measurements, offline sampling, LC-UV analysis, performed data analysis, and lead the writing of the manuscript. H. Hellén designed the measurement campaign, conducted GC-MS measurements and data analysis, and commented the



manuscript. T. Tykkä assisted GC-MS measurements and data analysis and commented on the manuscript. V. Vakkari provided mixing layer height data, their description in the method part and commented the manuscript. D. Chen, M. Boy and P. Zhou performed model runs with

the help of D. Taipale and all commented the manuscript.

*Acknowledgements.*   The presented research has been funded by the Academy of Finland (Academy Research Fellowship, project no. 275608, Postdoctoral Research project no. 307957, and Centre of Excellence in Atmospheric Science, grant no. 272041). The authors thank Hannele Hakola for the continuous support. They also thank the staff at the SMEAR II station for their help, Dr. Petri Keronen for providing the data that we retrieved from SmartSMEAR, Dr. Jari Waldén for lending calibration standards and gas analyzers, and Anne-Mari Mäkelä for the

analysis of the canister samples. The authors also wish to acknowledge CSC IT Center for Science, Finland, for computational resources.





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

**Appendix A**





Table 2: Averages of individual compounds mixing ratios [ppt$_v$] and calculated OH reactivity, $R_{OH}$ [s$^{-1}$], and group $R_{OH}$ for the three periods studied with SOSAA. 'n.d.' means 'not detected' and 'n.m.' means 'not measured'.

| | 29 – 30 April | | | | 22 – 25 May | | | | 24 – 26 July | | | |
|---|---|---|---|---|---|---|---|---|---|---|---|---|
| | Mixing ratio [ppt$_v$] | | $R_{OH}$ [s$^{-1}$] | | Mixing ratio [ppt$_v$] | | $R_{OH}$ [s$^{-1}$] | | Mixing ratio [ppt$_v$] | | $R_{OH}$ [s$^{-1}$] | |
| | mean | (std) | mean | (std) | mean | (std) | mean | (std) | mean | (std) | mean | (std) |
| *Alkanes* | | | *0.065* | *(0.013)* | | | *0.00037* | *(0.00025)* | | | *0.00035* | *(0.00021)* |
| ethane | 2280 | (77) | 0.0115 | (0.0003) | n.m. | (-) | - | (-) | n.m. | (-) | - | (-) |
| propane | 578 | (81) | 0.014 | (0.002) | n.m. | (-) | - | (-) | n.m. | (-) | - | (-) |
| n-butane | 140 | (49) | 0.0078 | (0.0027) | n.m. | (-) | - | (-) | n.m. | (-) | - | (-) |
| 2-methylpropane | 89 | (32) | 0.0045 | (0.0016) | n.m. | (-) | - | (-) | n.m. | (-) | - | (-) |
| n-pentane | 61 | (13) | 0.0055 | (0.0012) | n.m. | (-) | - | (-) | n.m. | (-) | - | (-) |
| 2-methylbutane | 113 | (16) | 0.011 | (0.002) | n.m. | (-) | - | (-) | n.m. | (-) | - | (-) |
| n-hexane | 23 | (8) | 0.0028 | (0.0009) | n.m. | (-) | - | (-) | n.m. | (-) | - | (-) |
| 2-methylpentane | 26 | (7) | 0.0035 | (0.0010) | n.m. | (-) | - | (-) | n.m. | (-) | - | (-) |
| n-heptane | 5.7 | (1.8) | 0.0020 | (0.0007) | 0.57 | (0.35) | 0.000064 | (0.000115) | 0.22 | (0.15) | 0.0000094 | (0.0000301) |
| n-octane | 7.3 | (2.6) | 0.0015 | (0.0005) | 1.0 | (0.4) | 0.00020 | (0.00009) | 1.9 | (0.4) | 0.00032 | (0.00016) |
| n-nonane | 3.3 | (1.3) | 0.00082 | (0.00031) | 0.44 | (0.15) | 0.00010 | (0.00004) | 0.13 | (0.07) | 0.000024 | (0.000019) |
| n-decane | 2.4 | (1.1) | 0.00066 | (0.00031) | n.d. | (-) | - | (-) | n.d. | (-) | - | (-) |
| *Alkenes* | | | *0.38* | *(0.04)* | | | - | (-) | | | - | (-) |
| ethene | 354 | (26) | 0.077 | (0.006) | n.m. | (-) | - | (-) | n.m. | (-) | - | (-) |
| propene | 135 | (6) | 0.11 | (0.01) | n.m. | (-) | - | (-) | n.m. | (-) | - | (-) |
| 1-butene | 47 | (4) | 0.043 | (0.004) | n.m. | (-) | - | (-) | n.m. | (-) | - | (-) |
| trans-2-butene | 46 | (8) | 0.087 | (0.015) | n.m. | (-) | - | (-) | n.m. | (-) | - | (-) |
| cis-2-butene | 27 | (4) | 0.043 | (0.007) | n.m. | (-) | - | (-) | n.m. | (-) | - | (-) |
| 1,3-butadiene | n.d. | (-) | - | (-) | n.m. | (-) | - | (-) | n.m. | (-) | - | (-) |
| 1-pentene | 35 | (7) | 0.025 | (0.005) | n.m. | (-) | - | (-) | n.m. | (-) | - | (-) |
| trans-2-pentene | n.d. | (-) | - | (-) | n.m. | (-) | - | (-) | n.m. | (-) | - | (-) |
| ethyne | 260 | (22) | 0.0052 | (0.0005) | n.m. | (-) | - | (-) | n.m. | (-) | - | (-) |
| *Aromatics* | | | *0.038* | *(0.014)* | | | *0.013* | *(0.009)* | | | *0.020* | *(0.017)* |
| benzene | 93 | (16) | 0.0028 | (0.0005) | 12 | (3) | 0.00035 | (0.00010) | 14 | (4) | 0.00036 | (0.00020) |
| toluene | 37 | (9) | 0.0058 | (0.0014) | 34 | (10) | 0.0048 | (0.0017) | 22 | (6) | 0.0026 | (0.0014) |
| ethylbenzene | 10 | (2) | 0.0018 | (0.0004) | 2.8 | (0.8) | 0.00047 | (0.00016) | 6.4 | (1.7) | 0.00092 | (0.00049) |
| p/m-xylene | 14 | (7) | 0.0067 | (0.0034) | 3.2 | (2.3) | 0.0015 | (0.0011) | 11 | (2) | 0.0041 | (0.0020) |
| o-xylene | 5.8 | (2.0) | 0.0020 | (0.0007) | 1.0 | (1.0) | 0.00019 | (0.00030) | 2.2 | (1.0) | 0.00062 | (0.00040) |
| styrene | 7.7 | (3.3) | 0.012 | (0.005) | 1.7 | (1.3) | 0.0024 | (0.0019) | 8.6 | (8.0) | 0.010 | (0.011) |
| 2-ethyltoluene | 1.2 | (0.3) | 0.00036 | (0.00010) | 0.44 | (0.23) | 0.000058 | (0.000080) | 0.61 | (0.24) | 0.00015 | (0.00009) |
| 3-ethyltoluene | n.d. | (-) | - | (-) | 0.88 | (1.61) | 0.00034 | (0.00069) | 0.27 | (0.15) | 0.00010 | (0.00008) |
| 4-ethyltoluene | 0.12 | (0.05) | 0.000036 | (0.000016) | 0.27 | (0.12) | 0.000034 | (0.000047) | 0.34 | (0.28) | 0.000031 | (0.000065) |





| | 29 – 30 April | | | | 22 – 25 May | | | | 24 – 26 July | | | |
|---|---|---|---|---|---|---|---|---|---|---|---|---|
| | Mixing ratio [pptv] | | $R_{OH}$ [s$^{-1}$] | | Mixing ratio [pptv] | | $R_{OH}$ [s$^{-1}$] | | Mixing ratio [pptv] | | $R_{OH}$ [s$^{-1}$] | |
| | mean | (std) | mean | (std) | mean | (std) | mean | (std) | mean | (std) | mean | (std) |
| 1,2,3-trimethylbenzene | 2.3 | (0.8) | 0.0019 | (0.0007) | 2.7 | (2.5) | 0.0021 | (0.0020) | 1.4 | (0.5) | 0.0010 | (0.0006) |
| 1,2,4-trimethylbenzene | 3.4 | (1.0) | 0.0028 | (0.0009) | 0.59 | (0.53) | 0.00031 | (0.0004) | 0.43 | (0.20) | 0.00028 | (0.00020) |
| 1,3,5-trimethylbenzene | 1.3 | (0.7) | 0.0021 | (0.0011) | 0.39 | (0.21) | 0.00020 | (0.0003) | 0.24 | (0.10) | 0.00024 | (0.00019) |
| isoprene | 4.1 | (3.0) | 0.011 | (0.009) | 8.5 | (6.7) | 0.022 | (0.017) | 29 | (14) | 0.059 | (0.040) |
| *Monoterpenoids* | | | *0.55* | *(0.40)* | | | *0.51* | *(0.55)* | | | *1.8* | *(1.4)* |
| α-pinene | 228 | (147) | 0.34 | (0.23) | 131 | (138) | 0.18 | (0.19) | 648 | (318) | 0.72 | (0.52) |
| β-pinene | 28 | (27) | 0.061 | (0.059) | 27 | (28) | 0.052 | (0.055) | 108 | (74) | 0.17 | (0.15) |
| camphene | 23 | (16) | 0.031 | (0.023) | 32 | (29) | 0.041 | (0.039) | 54 | (33) | 0.058 | (0.047) |
| Δ³-carene | 46 | (37) | 0.11 | (0.08) | 80 | (85) | 0.17 | (0.19) | 230 | (142) | 0.41 | (0.34) |
| p-cymene | 5.5 | (2.4) | 0.0021 | (0.0009) | 25 | (25) | 0.0089 | (0.0089) | 11 | (5) | 0.0033 | (0.0022) |
| limonene | 1.8 | (1.4) | 0.0040 | (0.0063) | 14 | (14) | 0.056 | (0.059) | 100 | (65) | 0.34 | (0.29) |
| terpinolene | n.d. | (-) | - | | 0.55 | (0.36) | 0.00078 | (0.00169) | 11 | (7) | 0.051 | (0.041) |
| myrcene | 0.26 | (0.26) | 1.6e-12 | (2.0e-12) | 4.4 | (3.0) | 2.0e-12 | (2.3e-11) | 14 | (9) | 8.2e-11 | (6.7e-11) |
| 1,8-cineol | 2.7 | (2.4) | 0.00079 | (0.00070) | 13 | (9) | 0.0035 | (0.0025) | 22 | (9) | 0.0050 | (0.0033) |
| bornylacetate | 0.31 | (0.21) | 0.00011 | (0.00008) | 1.8 | (0.9) | 0.00037 | (0.00040) | 2.7 | (1.3) | 0.00077 | (0.00053) |
| *Sequiterpenes* | | | *0.0012* | *(0.0027)* | | | *0.024* | *(0.025)* | | | *0.17* | *(0.14)* |
| longicyclene | 0.33 | (0.27) | 0.000081 | (0.000067) | 0.84 | (0.25) | 0.00010 | (0.00011) | 0.79 | (0.36) | 0.00015 | (0.00010) |
| iso-longifolene | 0.0600 | (0.0003) | 0.000046 | (0.000070) | 0.28 | (0.13) | 0.000042 | (0.000180) | n.d. | (-) | - | (-) |
| β-farnesene | n.d. | (-) | - | (-) | n.d. | (-) | - | | 4.0 | (1.4) | 0.014 | (0.008) |
| β-caryophyllene | 0.83 | (0.58) | 0.0010 | (0.0024) | 7.6 | (3.7) | 0.023 | (0.024) | 28 | (16) | 0.12 | (0.09) |
| α-humulene | 0.0514 | (0.0001) | 0.000070 | (0.000148) | 0.19 | (0.13) | 0.0013 | (0.0010) | n.d. | (-) | - | (-) |
| SQT1* | n.d. | (-) | - | (-) | n.d. | (-) | - | (-) | 2.8 | (1.5) | 0.0057 | (0.0042) |
| SQT2* | n.d. | (-) | - | (-) | n.d. | (-) | - | (-) | 5.5 | (3.3) | 0.011 | (0.009) |
| SQT3* | n.d. | (-) | - | (-) | n.d. | (-) | - | (-) | 4.5 | (2.4) | 0.0074 | (0.0072) |
| SQT4* | n.d. | (-) | - | (-) | n.d. | (-) | - | (-) | 12 | (6) | 0.018 | (0.018) |
| *GLVs* | | | - | (-) | | | *0.0022* | *(0.0021)* | | | *0.013* | *(0.020)* |
| 1-hexanol | n.d. | (-) | - | (-) | n.d. | (-) | - | (-) | 8.0 | (4.3) | 0.0010 | (0.0017) |
| cis-2-hexen-1-ol | n.d. | (-) | - | (-) | n.d. | (-) | - | (-) | n.d. | (-) | - | (-) |
| trans-2-hexen-1-ol | n.d. | (-) | - | (-) | n.d. | (-) | - | (-) | n.d. | (-) | - | (-) |
| cis-3-hexen-1-ol | n.d. | (-) | - | (-) | n.d. | (-) | - | (-) | 4.4 | (2.3) | 0.0027 | (0.0058) |
| trans-3-hexen-1-ol | n.d. | (-) | - | (-) | n.d. | (-) | - | (-) | 6.7 | (1.5) | 0.0064 | (0.0097) |
| trans-2-hexenal | n.d. | (-) | - | (-) | 2.6 | (1.8) | 0.0022 | (0.0021) | 3.4 | (2.2) | 0.0027 | (0.0026) |
| hexylacetate | n.d. | (-) | - | (-) | n.d. | (-) | - | (-) | n.d. | (-) | - | (-) |
| cis-3-hexenylacetate | n.d. | (-) | - | (-) | n.d. | (-) | - | (-) | n.d. | (-) | - | (-) |
| trans-2-hexenyl-acetate | n.d. | (-) | - | (-) | n.d. | (-) | - | (-) | n.d. | (-) | - | (-) |
| *Aldehydes* | | | *0.01* | *(0.07)* | | | *0.074* | *(0.052)* | | | *0.35* | *(0.12)* |





| | 29 – 30 April | | | | 22 – 25 May | | | | 24 – 26 July | | | |
|---|---|---|---|---|---|---|---|---|---|---|---|---|
| | Mixing ratio [pptv] | | $R_{OH}$ [s$^{-1}$] | | Mixing ratio [pptv] | | $R_{OH}$ [s$^{-1}$] | | Mixing ratio [pptv] | | $R_{OH}$ [s$^{-1}$] | |
| | mean | (std) | mean | (std) | mean | (std) | mean | (std) | mean | (std) | mean | (std) |
| formaldehyde | 127 | (111) | 0.029 | (0.026) | n.m. | (-) | - | - | 621 | (89) | 0.13 | (0.02) |
| acetaldehyde | 16.5 | (0.1) | 0.0016 | (0.0029) | n.m. | (-) | - | - | 343 | (62) | 0.13 | (0.02) |
| propanal | 83 | (30) | 0.045 | (0.016) | 90 | (47) | 0.038 | (0.026) | 111 | (35) | 0.017 | (0.027) |
| butanal | n.d. | (-) | - | (-) | 4.7 | (1.5) | 0.00045 | (0.00111) | 17 | (26) | 0.0085 | (0.0147) |
| pentanal | 19 | (6) | 0.015 | (0.005) | 25 | (21) | 0.012 | (0.015) | 41 | (16) | 0.028 | (0.011) |
| hexanal | 8.03 | (0.04) | 0.0016 | (0.0028) | 7.6 | (3.4) | 0.0054 | (0.0026) | 17 | (8) | 0.012 | (0.005) |
| heptanal | n.d. | (-) | - | | 5.9 | (1.6) | 0.0043 | (0.0013) | 0.07 | (0.08) | 0.0000010 | (0.0000105) |
| octanal | n.d. | (-) | - | | 4.2 | (1.0) | 0.0032 | (0.0009) | 6.1 | (1.6) | 0.0039 | (0.0021) |
| nonanal | n.d. | (-) | - | | 2.8 | (1.0) | 0.0024 | (0.0010) | 9.2 | (4.1) | 0.0068 | (0.0045) |
| decanal | n.d. | (-) | - | (-) | 3.1 | (0.8) | 0.0026 | (0.0008) | 9.8 | (3.0) | 0.0069 | (0.0039) |
| methacrolein | 8.0 | (1.5) | 0.0029 | (0.0033) | 8.2 | (3.5) | 0.0059 | (0.0026) | 7.1 | (6.7) | 0.0044 | (0.0047) |
| crotonaldehyde | 1.6 | (0.1) | 0.00082 | (0.00071) | n.m. | (-) | - | (-) | n.d. | (-) | - | (-) |
| benzaldehyde | 26 | (2) | 0.0017 | (0.0036) | n.m. | (-) | - | (-) | n.d. | (-) | - | (-) |
| tolualdehyde | 75 | (7) | 0.0063 | (0.0132) | n.m. | (-) | - | (-) | n.d. | (-) | - | (-) |
| *Alcohols* | | | *0.082* | *(0.079)* | | | *0.18* | *(0.52)* | | | *0.048* | *(0.119)* |
| isopropanol | 25 | (6) | 0.0035 | (0.0008) | 39 | (31) | 0.0043 | (0.0042) | 171 | (96) | 0.0070 | (0.0122) |
| 1-butanol | 347 | (348) | 0.079 | (0.079) | 969 | (2558) | 0.18 | (0.52) | 574 | (724) | 0.039 | (0.102) |
| 1-pentanol | n.d. | (-) | - | | 3.9 | (1.4) | 0.000073 | (0.000303) | 8.7 | (3.3) | 0.00078 | (0.00131) |
| 1-penten-3-ol | n.d. | (-) | - | | 1.9 | (0.7) | 0.00047 | (0.00128) | 3.6 | (2.3) | 0.0016 | (0.0035) |
| 3-methyl-2-buten-1-ol | n.d. | (-) | - | | n.d. | (-) | - | | n.d. | (-) | - | |
| 1-octen-3-ol | n.d. | (-) | - | | n.d. | (-) | - | | 1.5 | (0.4) | 0.00025 | (0.00058) |
| 2-methyl-3-buten-2-ol (MBO) | 5.7 | (4.7) | 0.022 | (0.019) | 16 | (16) | 0.059 | (0.060) | 47 | (28) | 0.15 | (0.12) |
| *Other carbonyls* | | | *0.013* | *(0.018)* | | | *0.0013* | *(0.0021)* | | | *0.050* | *(0.022)* |
| acetone (and acrolein) | 2798 | (3996) | 0.011 | (0.016) | n.m. | (-) | - | | 9176 | (1588) | 0.038 | (0.007) |
| 6-methyl-2-hepten-3-one | n.d. | (-) | - | | n.d. | (-) | - | | 1.5 | (0.6) | 0.00015 | (0.00063) |
| methyl ethyl ketone (MEK) | n.d. | (-) | - | | n.d. | (-) | - | | 9.0 | (0.3) | 0.00023 | (0.00005) |
| butylacetate | 2.8 | (1.3) | 0.000051 | (0.000143) | n.d. | (-) | - | | n.d. | (-) | - | (-) |
| 4-acetyl-1-methylcyclohexene | n.d. | (-) | - | | 1.3 | (0.6) | 0.00026 | (0.00112) | 4.5 | (4.1) | 0.0085 | (0.0123) |
| nopinone | 4.7 | (3.2) | 0.0017 | (0.0012) | 3.1 | (2.6) | 0.0010 | (0.0009) | 10 | (4) | 0.0030 | (0.0019) |
| *Organic acids* | | | *0.071* | *(0.013)* | | | *0.023* | *(0.018)* | | | *0.030* | *(0.045)* |
| acetic acid | 2796 | (459) | 0.060 | (0.009) | 1495 | (423) | 0.019 | (0.014) | 3016 | (287) | 0.018 | (0.026) |
| propanoic acid | 140 | (25) | 0.0044 | (0.0008) | 83 | (27) | 0.0017 | (0.0013) | 160 | (53) | 0.0015 | (0.0024) |
| butanoic acid | 95 | (36) | 0.0045 | (0.0017) | 55 | (31) | 0.0017 | (0.0016) | 153 | (26) | 0.0022 | (0.0032) |
| isobutanoic acid | n.d. | (-) | - | (-) | n.d. | (-) | - | | 33 | (15) | 0.00029 | (0.00070) |
| pentanoic acid | 20 | (13) | 0.0016 | (0.0015) | 21 | (11) | 0.00057 | (0.00114) | 178 | (35) | 0.0058 | (0.0087) |
| isopentanoic acid | 1.0 | (0.2) | 0.0000015 | (0.0000128) | 2.0 | (0.5) | 0.0000029 | (0.0000252) | 8.7 | (5.2) | 0.00023 | (0.00047) |

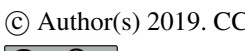



Atmospheric Chemistry and Physics Discussions — Open Access

| | 29 – 30 April | | | | 22 – 25 May | | | | 24 – 26 July | | | |
|---|---|---|---|---|---|---|---|---|---|---|---|---|
| | Mixing ratio [ppt$_v$] | | $R_{\mathrm{OH}}$ [s$^{-1}$] | | Mixing ratio [ppt$_v$] | | $R_{\mathrm{OH}}$ [s$^{-1}$] | | Mixing ratio [ppt$_v$] | | $R_{\mathrm{OH}}$ [s$^{-1}$] | |
| | mean | (std) | mean | (std) | mean | (std) | mean | (std) | mean | (std) | mean | (std) |
| 4-methylpentanoic acid | n.d. | (-) | - | (-) | n.d. | (-) | - | (-) | n.d. | (-) | - | (-) |
| hexanoic acid | 5.5 | (2.1) | 0.000042 | (0.000191) | 9.7 | (2.8) | 0.000063 | (0.000295) | 35 | (11) | 0.0015 | (0.0024) |
| heptanoic acid | n.d. | (-) | - | (-) | n.d. | (-) | - | (-) | 13 | (3) | 0.00052 | (0.00097) |
| *Inorganics* | | | 1.2 | (0.2) | | | 1.0 | (0.2) | | | 1.1 | (0.2) |
| NO | 77 | (41) | 0.013 | (0.012) | 91 | (54) | 0.013 | (0.014) | 69 | (42) | 0.010 | (0.010) |
| NO$_2$ | 460 | (387) | 0.14 | (0.13) | 428 | (312) | 0.11 | (0.09) | 152 | (94) | 0.034 | (0.027) |
| O$_3$ | 4.25e4 | (9.1e3) | 0.066 | (0.015) | 4.1e4 | (7.4e3) | 0.068 | (0.014) | 2.7e4 | (8.9e3) | 0.046 | (0.016) |
| SO$_2$ | 55 | (62) | 0.00091 | (0.00137) | 37 | (25) | 0.00056 | (0.00056) | 75 | (117) | 0.0012 | (0.0022) |
| CO | 1.29e5 | (9.4e3) | 0.72 | (0.06) | 1.1e5 | (6.1e3) | 0.58 | (0.03) | 1.4e5 | (2.0e4) | 0.73 | (0.11) |
| CH$_4$ | 1.9e6 | (2.1e3) | 0.22 | (0.01) | 1.9e6 | (5.2e3) | 0.26 | (0.02) | 1.9e6 | (2.2e-16) | 0.28 | (0.01) |
| *Model OVOCs* | | | 0.12 | (0.06) | | | 0.25 | (0.12) | | | 0.78 | (0.36) |
| *Model inorganics* | | | 0.057 | (0.004) | | | 0.070 | (0.007) | | | 0.077 | (0.005) |
| *Missing* | | | 11 | (13) | | | 6.4 | (5.6) | | | 8.6 | (5.3) |
| *Total* | | | 13.3 | (13.7) | | | 8.6 | (5.9) | | | 13.0 | (5.9) |

* quantified with $\beta$-caryophyllene calibration and an estimated reaction coefficient (1e-10 cm$^3$ s$^{-1}$)

640