# Peer review of "Long-term total OH reactivity measurements in a boreal forest"

_Atmospheric Chemistry and Physics, 2019_

## Referee Comment (RC1) · Anonymous Referee #1 · 18 Mar 2019

The authors report OH reactivity measurements at a measurement site in Finland, where a large fraction of measured OH reactivity was not explained in previous campaigns. Here, the authors measured a larger set of OH reactants for at least part of the measurement period. Nevertheless, similar results as in previous studies were found demonstrating that the nature of a large fraction of OH reactants in the Boreal forest in Finland is not known. The manuscript is well suited for publication in ACP after considering the following points:

The manuscript could be more focussed on the measurement and less on the instrumental part. A large fraction of the instrumental part is repeating what is described in an earlier paper by the authors. Figures are in general very complex and contain a lot of information. I would suggest re-considering, if all information is needed and what

can be taken out or moved to the Appendix.

P5 l11: I think it should read something like: "The OH reactivity of a compound is the inverse lifetime of OH with respect to its reaction with the compound."

P5 l23: The authors might want to mention that HO2 is concurrently produced.

P6 l11: Better give quantitative numbers instead of a qualitative statement "usually small"

P6 l27: There is something missing in the reference

P7 l3: "assumes" instead of "assume"

P7 l4/5: The authors might want to consider rephrasing the sentence.

P7 l8-10: First, the authors state that the correction was applied for certain conditions, but say in the next sentence that conditions were never met for the correction. I would suggest combining the statements.

P11 l12/13: I would suggest explaining what kind of "amendments by Michoud" were included. What is meant by the "minor improvements"?

P12 l10: What is meant by "photochemistry has been improved"? What are the changes and how important are they?

P12 l16: It would be helpful for the reader to get an estimate of the lifetime of oxygenated VOCs in the model, in order to judge how important deposition was.

P14 Table 1: Please indicate what x and y in the regression is.

P14 l4: The authors mention an exponential fit, but show a linear fit in Table 1. I would rather give one approach.

P14 l7/8: Is this statement justified? This is also the period, when the lowest number of instruments measured OH reactants.

[Figure]

P18 l8: Is there a hint that measured OVOCs are not explained by gas-phase oxidation, but require such re-emission processes to justify this hypothesis?

P18 l17: "indicate" instead of "indicates"

P20: The discussion about the additional uncertainty of the O3 correction from the O3 measurement being at a different sampling point might better fit earlier in the instrumental section.

Figures in general: Symbols are often too small and hard to distinguish. Font sizes of legend texts are often too small.
* * *

---

## Referee Comment (RC2) · Anonymous Referee #2 · 29 Apr 2019

General Comments: The paper by Praplan et al. presents total OH reactivity measurements alongwith OH reactant measurements carried out in a boreal forest environment during the months of April, May, June and July in 2016. Measurements of total OH reactivity over a four-month period are indeed rare at any atmospheric environment and without doubt the suite of VOC measurements are impressive. However, there are several issues which I think the authors need to address through substantive revisions. Major Concerns: 1) Experimental methods: The description of the OH reactivity measurement method with the multiple corrections listed by the authors (only to later surmise that many of those corrections were not necessary for their ambient conditions is confusing and at times also misleading. Though they cite the original method paper published in Atm Env (Praplan et al. 2017) which was set up for urban measurements

in high NO environment, I have to say that there are several issues pertaining to their treatment of the details. The authors seem to make strange assumptions concerning the competitive kinetics occurring inside the reactor. While they consider the potential for OH production inside the reactor through O3 and NO2 photolysis as well as NO + HO2 reactions, I could not find any discussion of compensating effects in such reactions. Firstly, the C1-C2 signal is a direct measurement of the OH in the reactor available for competitive reactions during the C3 stage when ambient air is sampled and this already takes into account ozone formed from the UV lamp in the absence of ambient air being sampled. When ambient air containing 50-60 ppb of ozone is sampled into the reactor it also gets diluted so the moot question really is how much OH do the authors think can be produced near the reaction zone which is just where photons from the lamp and the ambient air reactants mix in. The lifetime of OH radicals just against pyrrole molecules is few milli-seconds and while the lamp position will make some difference, I wonder how much additional OH the authors think can come from the Ozone photolysis when those same photons have about 3 times higher number of pyrrole molecules and several times higher number of water molecules to compete against?? Can they show a simple calculation for all these relative channels? Now coming to the NOx reactions the authors also do need to consider compensatory reactions that can mitigate the OH reformation as well which they have simply ignored...for example when NO2 photolyzes to NO, they should also consider that the 100's of ppb of ozone inside the reactor will convert the NO back to NO2 really fast as well. The additional NO2 can be a sink of the additional OH, thereby cancelling some of the extra "OH" effect. While NOx may not be important for the forest measurements, the general point is that such compensatory reactions can have a "benign" influence and before launching "detailed" box model simulations focusing on only a subset of reactions to understand the chemistry inside the CRM, the authors need to consider such aspects more thoroughly (both in the 2017 work and this work). For each of the interferences they mentioned a simple set of experiments with varying NOx or O3 or humidity in the reactor with the probe sampling the air into the GC-FID from the reactor at different

distances from the lamp would have revealed more on how significant these effects could turn out to be. The authors do acknowledge that the chemistry inside the reactor and the box model analysis of the chemistry inside the reactor are not completely understood. . .in such case it would have been better to rely on experimental calibrations which better validate the method then to rely on more uncertain model corrections just because these models give the sense of being "detailed". So my suggestion is that the authors stick to only those interferences which are relevant for the forest environment measurements in their study and do a thorough job of addressing them also considering compensatory effects in the revised version. It would also be nice to know whether authors tested the GC FID signal for humidity interferences of pyrrole detection. The concept paper by Noelscher et al. 2012 did mention this as a GC-FID "detector " specific issue for CRM measurements.. How often during the four month deployment was the sensitivity drift in the GC-FID characterized through calibrations and corrected for ? How often were CRM calibration experiments performed in the four month period? Were there any major changes in the detection sensitivity ? This is important to address as the June data seems to be quite dissimilar relative to other months despite no obvious changes in the co-measured OH reactants.

Corrections for deviation from pseudo first order conditions: The authors mention quite a lot about this but I could not find exactly what molecule they used as a proxy in their simulations to determine the correction factor. Was it propane? If so, then this is not the ideal choice for a forested environment where terpenes form a major fraction of the ambient reaction mixture as typically terpenes are 10-100 times faster on per molecule basis with the OH radical. The correction factor depends quite strongly on the choice of the molecule with higher correction factors for propane and lower/no correction required for a terpene compound like isoprene or alpha –pinene. It would be good to have some discussion of this sensitivity as it has a direct bearing on the measured and missing OH reactivity calculations. . . 2) Box model for understanding chemistry inside CRM: This is the part that I found to be scant on details and sensitivity runs . . .I had little confidence in this analysis after reading the scanty description of how

the box model was set up and the tenuous statements made on its basis for example didn't make me understand it any better: "This box model is far from taking into account the complex processes in the CRM reactor, but it is a useful tool to test hypotheses (such as NO2 and O3 photolysis) and to extend the validity range of correction factors that depend on pyr:OH to conditions that were not available experimentally..." I would suggest complete removal of this part of the analysis unless the authors can refine it and also show what useful and critical info came out of this... Given the published literature on CRM , the experimental corrections listed in Noelscher et al. 2012 and the original Sinha et al. 2008 paper are sufficient for this study.

3) Interpretation of ambient measurements: Most of the figures showing the ambient data were not easy to read and seemed to be too cluttered. I request the authors to improve the figures...for e.g. there is hardly any need to show multiple traces of measured OH reactivity with different stages of corrections ..they can list these magnitude ranges in the experimental section and show the final values they used... I also found the June data quite surprising (Table 1). Measured MT were lower than May but measured total OH reactivity was highest as was the fraction of missing reactivity (which the authors attribute to lack of co-measured OH reactants). Surely there may be few days of data in June that have better coverage which can be used instead of the monthly average? The authors mention alkyl amines as a source from soil. Can they rule out the contribution of biomass fires during the four months, esp in June? Kumar et al., 2018 Sci Reports have reported amides and amine in biomass burning plumes while trying to explain their missing OH reactivity and the authors may want to check out this possibility. Also it is surprising that no PTR-MS measurements from the SMEAR station were included in the analyses. Online measurements at high temporal resolution of acetonitrile or acetaldehyde would have bene useful but perhaps they can still look at the CO data if PTR-MS measurements were unavailable? Finally some comment is warranted to justify the SOSAA model comparison with the measurements made at very different heights from the height at which the model is simulating the chemistry.....can't vertical gradients confound such comparisons?

I hope that with the above revisions that address the major points raised above, the paper can become suitable for publication in ACP as the novelty of the work and the need for such data is high.
* * *

---

## Author Comment (AC1) · 7 Jun 2019

**Answers to the referees**

We thank the referees for their reviews. We provide here some answers and mention the changes that we made to our manuscript to address the referees' concerns and remarks. Referees' comments are in italics.

**Anonymous referee #1**

*The authors report OH reactivity measurements at a measurement site in Finland, where a large fraction of measured OH reactivity was not explained in previous campaigns. Here, the authors measured a larger set of OH reactants for at least part of the measurement period. Nevertheless, similar results as in previous studies were found demonstrating that the nature of a large fraction of OH reactants in the Boreal forest in Finland is not known. The manuscript is well suited for publication in ACP after considering the following points:*

*The manuscript could be more focussed on the measurement and less on the instrumental part. A large fraction of the instrumental part is repeating what is described in an earlier paper by the authors. Figures are in general very complex and contain a lot of information. I would suggest reconsidering, if all information is needed and what can be taken out or moved to the Appendix.*

Our intention was to provide enough background information on the measurement method and underlying assumptions to allow for a meaningful and comprehensive discussion and interpretation of the results. We apparently did not manage to find the right balance between these two parts. Taking also into account the comments from Anonymous referee #2, we decided to approach the corrections of the CRM data without relying on the box model for the CRM reactor (previously 2.6.1) and removed this section. As a consequence, the experimental part is now streamlined and should be clearer to follow and technical details about these correction factors have been moved to the Appendix, following the referee's suggestion. We added information that was missing according to both referees' comments, tried to avoid digression and did our best to remain concise.

*P5 l11: I think it should read something like: "The OH reactivity of a compound is the inverse lifetime of OH with respect to its reaction with the compound."*

We agree with the referee that our statement is inaccurate. It has been changed to "The OH reactivity of a compound is the inverse of the OH chemical lifetime due to its reaction with that compound".

*P5 l23: The authors might want to mention that HO2 is concurrently produced.*

We now mention this in the text. ("Note that hyroperoxy radicals ($HO_2$) are concurrently produced from the reaction of hydrogen (H) with molecular oxygen ($O_2$).")

*P6 l11:  Better give quantitative numbers instead of a qualitative statement "usually small"*

We replaced "usually small" with the following quantitative statement: "4% or less 99% of the time, which corresponds to a change of no more than 5% for $R_{eqn}$".

*P6 l27: There is something missing in the reference*

We fixed the reference.

*P7 l3: "assumes" instead of "assume"*

This sentence has been removed from the revised manuscript.

*P7 l4/5: The authors might want to consider rephrasing the sentence.*

We have removed this sentence (see below).

*P7 l8-10: First, the authors state that the correction was applied for certain conditions, but say in the next sentence that conditions were never met for the correction. I would suggest combining the statements.*

In light of the previous comments and the streamlining of the experimental part, we do not refer anymore to correction factors due to the presence of $NO_x$, but only to the correction factor due to the presence of $NO_2$ and briefly acknowledge the low NO concentrations in the main section about the CRM.

*P11 l12/13: I would suggest explaining what kind of "amendments by Michoud" were included. What is meant by the "minor improvements"?*

The amendments by Michoud et al. (2015) were namely the namely the reaction of H with $O_2$, the reaction of $HO_2$ with itself, reactions of $RO_2$ produced by the oxidation of pyrrole with $RO_2$, $HO_2$ and NO (producing RO), and the reaction of RO with $O_2$. Minor improvements were enumerated in the original manuscript ("varying temperature, pressure and RH"). We called them minor as their influence on the results was negligible. However, due to the removal of this section, this is now irrelevant to the revised manuscript.

*P12 l10: What is meant by "photochemistry has been improved"? What are the changes and how important are they?*

We wrote in the original manuscript "by calculating the photodissociation constants more precisely using data from (Atkinson et al., 1992)". The photochemistry in SOSAA has been validated by calculating the photodissociation constants more precisely when using data from Atkinson et al. (1992) compared to the simplified version normally applied in MCM. This change has been done in Mogensen et al. (2011) and it is now mentioned explicitly in the revised manuscript. Since then several published studies showed that the new calculations of the photodissociation constants has improved the photochemistry of SOSAA (e.g. Boy et al., 2013).

*P12 l16: It would be helpful for the reader to get an estimate of the lifetime of oxygenated VOCs in the model, in order to judge how important deposition was.*

With the data of deposition velocities ($vd_i$) for each layer (calculated in SOSAA for each compound), the deposition lifetime of a compound inside the canopy can be calculated as:

$$\tau = \Sigma([C]_i * dh_i) / \Sigma(vd_i * [C]_i).$$

Here, the sum ($\Sigma$) is the summation from bottom (level 2 where soil deposition is calculated; we do not count from level 1 because it is the ground boundary which is only used in a numerical computation sense) to the canopy top (level 20, 19.96 m). We show examples of lifetimes due to dry deposition of some compounds mentioned in paper by Zhou et al. (2017) for July 2010 at SMEAR II in Fig. A1. The monthly mean lifetime is shown in the plot legend. To clarify this for the reader of our manuscript we extended the last sentence of this paragraph, which is now:

"The latter describes the explicit simulation of the loss of every compound in the model by dry deposition inside the canopy for all height levels and shows that the sink by dry deposition inside the canopy is comparable to the chemical production for several oxidised VOCs (e.g. pinic acid or BCSOZOH - reaction product of beta-caryophyllene)."

[Figure]

Figure A1: Deposition lifetimes for selected compounds in SOSAA. Top: selected monoterpenes. Middle: selected oxidation product with low molar masses. Bottom: selected oxidation products with higher molar masses.

*P14 Table 1: Please indicate what x and y in the regression is.*

We replaced $x$ with $R_{exp}$ in Table 1 for clarification.

*P14 l4: The authors mention an exponential fit, but show a linear fit in Table 1. I would rather give one approach.*

Our aim was to discuss the specific relationship between emissions and temperature, but we removed this now for the sake of clarity and concision.

*P14 l7/8: Is this statement justified? This is also the period, when the lowest number of instruments measured OH reactants.*

The reviewer is entirely right that this statement should have referred to the lack of data available to derived $R_{OH}$. However, due to the aforementioned changes in the analysis of CRM data, $R_{exp}$ and $R_{missing,fraction}$ are not both highest in June and we removed entirely this statement from the revised manuscript and revised the discussion.

*P18 l8: Is there a hint that measured OVOCs are not explained by gas-phase oxidation, but require such re-emission processes to justify this hypothesis?*

This re-emission effect is our hypothesis which, as far as we know, has never been considered explicitly in any current chemistry transport models. The emission models are usually based on measurement data, so we suppose this re-emission effect (if it exists) has already been included in the final empirical equations for both emission and deposition processes. So we decided to remove

this whole sentence ("The remaining missing reactivity could be also explained by oxidation products that were deposited and re-emitted from surfaces (so that they would not be taken into account when modelling their concentrations from atmospheric production based on their precursors concentrations.") and other references to that effect.

*P18 l17: "indicate" instead of "indicates"*

We fixed the typo.

*P20: The discussion about the additional uncertainty of the O3 correction from the O3measurement being at a different sampling point might better fit earlier in the instrumental section.*

The reason for placing this discussion was that it includes some results. However, we understand why it is better to bring it up earlier so that readers read the discussion section with this information in mind. Therefore, we moved this discussion at the end of the instrumental part in relation to the $O_3$ correction factor (section 2.5.2 of the revised manuscript).

*Figures in general: Symbols are often too small and hard to distinguish. Font sizes of legend texts are often too small.*

We reviewed the figures and made changes in order to increase their quality and readability.

***Anonymous Referee #2***

*General Comments: The paper by Praplan et al. presents total OH reactivity measurements along with OH reactant measurements carried out in a boreal forest environment during the months of April, May, June and July in 2016.  Measurements of total OH reactivity over a four-month period are indeed rare at any atmospheric environment and without doubt the suite of VOC measurements are impressive. However, there are several issues which I think the authors need to address through substantive revisions.*

*Major Concerns:*

*1) Experimental methods: The description of the OH reactivity measurement method with the multiple corrections listed by the authors (only to later surmise that many of those corrections were not necessary for their ambient conditions is confusing and at times also misleading. Though they cite the original method paper published in Atm Env (Praplan et al. 2017) which was set up for urban measurements in high NO environment, I have to say that there are several issues pertaining to their treatment of the details.*

We understand that both referee found some of the experimental part confusing, in particular regarding correction factors. Our original intent was to build on the modelling work from (Michoud et al., 2015), but we realise that in doing so we have made the paper less intelligible.

In the light of the comments from this referee, we decided that even though it is worth pursuing modelling of chemical processes in the CRM reactor, the present manuscript is not the best outlet for it and a dedicated manuscript with all the details would be a better option. Therefore, we removed the description of the box model for the CRM reactor and removed references to it. This means, that we rewrote completely the section dedicated to the pseudo first-order kinetic correction factor and removed references to model runs in the sections concerning other correction factors. Consequently, the data analysis of the CRM data has been modified according to the referee's

comments regarding correction factors. While absolute values changed, most of the main conclusions and discussion points from the paper still hold.

*The authors seem to make strange assumptions concerning the competitive kinetics occurring inside the reactor. While they consider the potential for OH production inside the reactor through O3 and NO2 photolysis as well as NO + HO2 reactions, I could not find any discussion of compensating effects in such reactions. Firstly, the C1-C2 signal is a direct measurement of the OH in the reactor available for competitive reactions during the C3 stage when ambient air is sampled and this already takes into account ozone formed from the UV lamp in the absence of ambient air being sampled.*

The $C_1$-$C_2$ signal is a direct measurement of the OH in the reactor available, but in zero air, which does not contain $NO_x$ and (additional) $O_3$ from sampled ambient air. While $O_3$ produced in the reactor from the UV lamp is indeed taken into account (see also reply below), the correction factors for NO, $NO_2$ (e.g. Michoud et al., 2015; Praplan et al., 2017) and $O_3$ (see Fuchs et al., 2017) adjust the OH availability from the measured $C_1$-$C_2$ signal in zero air to ambient conditions (similarly to changes in humidity between $C_2$ and $C_3$ states). It has been demonstrated experimentally that introducing these compounds to the reactor changes the OH production and consequently decreases the $C_3$ value (while $C_2$ remains the same), including in the present manuscript.

The box model used in the original manuscript contained the complete MCM scheme for inorganics as well as the amendments by Michoud et al. (2015) to take into account the peroxy radicals produced by the oxidation of pyrrole (Table S1 from Praplan et al., 2017)), so that compensating effects/reactions were considered. Nevertheless, as we removed the modelling of the CRM reactor from our analysis, this is not relevant anymore.

*When ambient air containing 50-60 ppb of ozone is sampled into the reactor it also gets diluted so the moot question really is how much OH do the authors think can be produced near the reaction zone which is just where photons from the lamp and the ambient air reactants mix in. The lifetime of OH radicals just against pyrrole molecules is few milliseconds and while the lamp position will make some difference, I wonder how much additional OH the authors think can come from the Ozone photolysis when those same photons have about 3 times higher number of pyrrole molecules and several times higher number of water molecules to compete against?? Can they show a simple calculation for all these relative channels?*

Is it indeed true that there is a large amount of $O_3$ in the reactor due to the UV lamp. We measured this value to be 170 $ppb_v$ as reported in our manuscript (p.11 l.17 of the original manuscript). With a dilution factor of about 1.4 and ambient $O_3$ concentrations of 50-60 $ppb_v$ it is still about 35-43 $ppb_v$ of $O_3$ that reach the reactor, which represents about a 20-25% increase of ozone in the reactor compared to the amount of ozone produced by the UV lamp. In our study pyrrole levels average about 32 $ppb_v$ and the $C_1$-$C_2$ signal vary between 5 and 30$ppb_v$, so that the decrease observed experimentally of up to about 3 $ppb_v$ (less than 10%) of the pyrrole signal is consistent with a photolysis of less than half of the introduced $O_3$ in the reactor, leading to additional formation of OH. If it is accepted that pyrrole photolyses at such concentrations (mostly at 254nm), then ozone should also be able to get photolysed (for wavelength smaller than 320nm). Our understanding is that water absorbs and photolyses to produce OH mostly at 185nm.

We acknowledge that we failed to mention explicitly in our original manuscript how much pyrrole is injected into our system. Therefore, we now mention explicitly in the experimental part that the amount of pyrrole injected in our setup between 26 and 43 $ppb_v$. As a result the pyrrole concentration is not about 3 times higher, but roughly the same as the additional ozone from ambient air.

*Now coming to the NOx reactions the authors also do need to consider compensatory reactions that can mitigate the OH reformation as well which they have simply ignored...for example when NO2 photolyzes to NO, they should also consider that the 100's of ppb of ozone inside the reactor will convert the NO back to NO2 really fast as well. The additional NO2 can be a sink of the additional OH, thereby cancelling some of the extra "OH" effect. While NOx may not be important for the forest measurements, the general point is that such compensatory reactions can have a "benign" influence and before launching "detailed" box model simulations focusing on only a subset of reactions to understand the chemistry inside the CRM, the authors need to consider such aspects more thoroughly (both in the 2017 work and this work).*

As we have not explicitly listed the reactions in the model, but only stated that we use the MCM scheme for inorganics with the amendments by Michoud et al. (2015), as mentioned previously, which represent 39 reactions (for inorganic species and pyrrole), which should include most chemical pathways. Even if, as we acknowledged in the original manuscript, the box model is not entirely accurate, it does take into account the roughly 170 $ppb_v$ of ozone present in the reactor and the conversion of NO back to $NO_2$. Our parametrization of the photolysis came from comparison of model runs with experimental data in order to take compensatory reactions into account. However, as we decided to follow the referee's recommendation below to stick to correction factors derived from experiments, this discussion is not relevant to the revised manuscript.

*For each of the interferences they mentioned a simple set of experiments with varying NOx or O3 or humidity in the reactor with the probe sampling the air into the GC-FID from the reactor at different distances from the lamp would have revealed more on how significant these effects could turn out to be.*

We deplore the fact that we could not make clear enough that we did exactly the experiments that the referee suggests and that we compared the results with our box model and used them to parametrize the photolysis. After doing so, we varied the conditions in the model to assess the effect of changes of the experimental conditions.

By removing the model box for the CRM reactor entirely and not comparing its results with experimental data, we expect the corresponding figures to be clearer and easier to interpret. We regret the confusion, but we are confident that the revised data analysis with correction factors based on these experimental results for $NO_2$ and $O_3$ injection in our system is easier to comprehend and addresses the referee's concerns.

*The authors do acknowledge that the chemistry inside the reactor and the box model analysis of the chemistry inside the reactor are not completely understood...in such case it would have been better to rely on experimental calibrations which better validate the method then to rely on more uncertain model corrections just because these models give the sense of being "detailed".*

We mention in the original manuscript (p.8 l.28-30) potential drawbacks of the experimental approach and made an attempt at a different model-based approach, by improving the work from Michoud et al. (2015). We think that our lack of clarity led to misunderstanding regarding our approach in the original manuscript.

This is another reason why we reconsidered our approach and opted for a purely experimental one in the revised manuscript. We see the benefit of focusing on the data using known corrections, keeping complicated modelling discussions for another publication, rather than cluttering the present manuscript with it.

*So my suggestion is that the authors stick to only those interferences which are relevant for the forest environment measurements in their study and do a thorough job of addressing them also considering compensatory effects in the revised version.*

As mentioned throughout our answers, taking into account both referees' comments regarding the experimental part of our manuscript, we have removed the part dedicated to the chemical modelling of the CRM reactor and redone the analysis with a different correction factors in order to keep the experimental section intelligible. We also placed some supporting data for the correction factors to the Appendix to keep the experimental section concise and as not to distract the reader from the field measurements. Absolute numbers changed as a result, but not the trends that we observed. We do hope that our answers and revised approach are addressing the referees' concerns in the best possible way.

*It would also be nice to know whether authors tested the GC-FID signal for humidity interferences of pyrrole detection. The concept paper by Noelscher et al. 2012 did mention this as a GC-FID "detector" specific issue for CRM measurements.*

We use a commercial GD-PID from Synspec BV (Groningen, The Netherlands), while Nölscher et al. (2012) used "a custom-built GC-PID system (VOC-Analyzer from IUTBerlin, now Environics-IUT GmbH)" so that a different behaviour can be expected. It is true that we did not mention in this study (or in our previous publication) that we checked for humidity interferences on the GC-PID signal and could not find any, in contrast to the concept paper by Nölscher et al. A test performed on 30 June 2015 with pyrrole calibrations performed at high and low RH, respectively, is shown in Fig. A2. Both calibration factors lie within the uncertainties of the measurements. We mention this explicitly in the revised version of the manuscript and added a figure.

[Figure]

Figure A2: Left: GC-PID sensitivity for pyrrole used in the present study. Right: Same day test (30 June 2015) for GC-PID sensitivity to pyrrole under dry and humid conditions.

*How often during the four month deployment was the sensitivity drift in the GC-FID characterized through calibrations and corrected for? How often were CRM calibration experiments performed in the four month period? Were there any major changes in the detection sensitivity? This is important to address as the June data seems to be quite dissimilar relative to other months despite no obvious changes in the co-measured OH reactants.*

The sensitivity of the GC-PID was measured at the beginning of the campaign (1645 a.u./ppb$_v$ on 5 April). This calibration was used for the data until June. The sensitivity was measured again on 30 July (1290 a.u./ppb$_v$) and this value was used for the data in July. We unfortunately have not measured the sensitivity between these two calibrations, but measurements of $C_0$ on 26 April and 31 May (34.7 ± 2.6 ppb$_v$ and 34.6 ± 0.5 ppb$_v$, respectively) do not reflect a loss in sensitivity.

The reason for higher OH reactivity values in June lie mostly in the correction factors as pyr:OH was very close to 1 during that period and it is a regime where correction factors become much larger. In the revised manuscript with the different approach to these corrections (in particular the pseudo first-order kinetics correction), the values are not as high as in the original manuscript.

*Corrections for deviation from pseudo first order conditions: The authors mention quite a lot about this but I could not find exactly what molecule they used as a proxy in their simulations to determine the correction factor. Was it propane? If so, then this is not the ideal choice for a forested environment where terpenes form a major fraction of the ambient reaction mixture as typically terpenes are 10-100 times faster on per molecule basis with the OH radical. The correction factor depends quite strongly on the choice of the molecule with higher correction factors for propane and lower/no correction required for a terpene compound like isoprene or alpha–pinene. It would be good to have some discussion of this sensitivity as it has a direct bearing on the measured and missing OH reactivity calculations…*

We attempted at providing a slightly different approach to the pseudo-first-order kinetics correction (combining experimental data and model runs). Our intention was to extend the conclusions of Michoud et al. (2015) by continuing the work they started with their simple model and use that as a framework for corrections. However, in order to address this referee's concerns, we moved away from this approach and used the same calibrations that we used to compare the model with in a purely experimental approach similar to the one from Sinha et al. (2008).

It is stated in the original manuscript (p. 8 l.31-32) that we used a commercial 10ppmv gas mixture for propane in $N_2$ and a home-made α-pinene in air standard which was analysed on several occasions by GC/MS in order to monitor the α-pinene concentrations and potential impurities.

We derived correction of the form $R_{CRM} = F_1 R_{eqn}^{F_2} + F_3$ (= $R_{true}$) according to numerical simulations described by Sinha et al. (2008) for various pyr:OH (Fig. A3), derived pyr:OH-dependent $F_1$, $F_2$, and $F_3$ values (Fig. A4), which we applied to the calibration data (Fig. A5). The slope of the propane calibration is consistent with results from Sinha et al. (2008), but we are using the α-pinene calibration to account for the difference between the real OH reactivity and the measured reactivity: $R_{measured} = (R_{CRM}+0.449)/0.497$

[Figure]

Figure A3: Numerical simulations of the relationship between Rtrue and Reqn for various pyr:OH values.

[Figure]

Figure A4: $F_1$, $F_2$, and $F_3$ dependence on pyr:OH.

[Figure]

Figure A5: Comparison between expected and measured OH reactivity values for propane (C3H8) and α-pinene calibrations.

*2) Box model for understanding chemistry inside CRM: This is the part that I found to be scant on details and sensitivity runs...I had little confidence in this analysis after reading the scanty description of how the box model was set up and the tenuous statements made on its basis for example didn't make me understand it any better: "This box model is far from taking into account the complex processes in the CRM reactor, but it is a useful tool to test hypotheses (such as NO2 and O3 photolysis) and to extend the validity range of correction factors that depend on pyr:OH to conditions that were not available experimentally..." I would suggest complete removal of this part of the analysis unless the authors can refine it and also show what useful and critical info came out of this...Given the published literature on CRM, the experimental corrections listed in Noelscher et al. 2012 and the original Sinha et al. 2008 paper are sufficient for this study.*

We appreciate the referee's criticism and acknowledge again that we did not successfully make the case for a different approach to corrections and that the description of our box model was insufficient. Instead of bloating the manuscript with much details and discussion about the model, we decided after careful consideration to redo the analysis according to the referee's suggestions in order to shift the focus to the actual field measurements, which are the important part of this study.

*3) Interpretation of ambient measurements:  Most of the figures showing the ambient data were not easy to read and seemed to be too cluttered. I request the authors to improve the figures... for e.g. there is hardly any need to show multiple traces of measured OH reactivity with different stages of corrections .. they can list these magnitude ranges in the experimental section and show the final values they used…*

In our attempt to present our results in the most comprehensive way, we recognise that we might have cluttered the figures. We decided originally to show two sets of data with various corrections in order to provide a visualization of the effect of this specific correction factor. At the referee's request, we show in the revised manuscript only the fully corrected data. We also opted for 1h averages for every time series as an attempt to make figures clearer. Finally we worked on a larger version of the plot for better readability (larger fonts) that is shown in a landscape orientation in the revised manuscript.

*I also found the June data quite surprising (Table 1).  Measured MT were lower than May but measured total OH reactivity was highest as was the fraction of missing reactivity (which the authors attribute to lack of co-measured OH reactants). Surely there maybe few days of data in June that have better coverage which can be used instead of the monthly average?*

In June the system was operating at Pyr:OH close to 1 (see above). In the revised revision of the manuscript, the OH reactivity is lower (and relatively constant) in June, which seems to be due to a cold spell during that time. The discussion has been modified accordingly.

*The authors mention alkyl amines as a source from soil. Can they rule out the contribution of biomass fires during the four months, esp. in June? Kumar et al., 2018 Sci Reports have reported amides and amine in biomass burning plumes while trying to explain their missing OH reactivity and the authors may want to check out this possibility.*

Occasionally, long-range transported biomass burning emissions are observed at the measurement site (Leino et al., 2014). During spring and summer 2016 biomass burning influence was low at Hyytiälä. Only between 23 and 26 July CO was elevated from 100 $ppb_v$ background level to approximately 150 $ppb_v$. This is still a low CO concentration compared to the biomass burning episodes analysed by Leino et al. (2014), when CO was larger than 250 $ppb_v$. We included this information in the discussion. Furthermore, alkyl amines have been measured from the forest floor in Hyytiälä (manuscript in preparation, Hemmilä et al.).

*Also it is surprising that no PTR-MS measurements from the SMEAR station were included in the analyses. Online measurements at high temporal resolution of acetonitrile or acetaldehyde would have bene useful but perhaps they can still look at the CO data if PTR-MS measurements were unavailable?*

CO data from the SMEAR station is included in the analysis as stated in the experimental part (section 2.1). No PTR-MS data were available during the time period of our measurements and we mention this explicitly now in the experimental part and the discussion. However, additional offline sampling performed between 27 April and 3 May and between 20 and 29 July gave average acetaldehyde values of 17 and 342 $ppt_v$, respectively, corresponding to OH reactivity values of 0.002 and 0.13 $s^{-1}$. This is consistent with typical values (from other years) measured at the site with PTR-MS. Typical concentrations for the less reactive acetonitrile are about 400 $ppt_v$ in average at the site (translating to roughly 0.0002 $s^{-1}$). These contributions to total OH reactivity remain small, but non-negligible for low OH reactivity values. We acknowledge this now explicitly in the discussion of the revised manuscript.

*Finally some comment is warranted to justify the SOSAA model comparison with the measurements made at very different heights from the height at which the model is simulating the chemistry.....can't vertical gradients confound such comparisons?*

We interpolated the model results to the measurement height for comparisons. In SOSAA set-up, the model layers closest to 1.5 m are level 6 (1.23 m) and level 7 (1.61 m). So the vertical gradient or the interpolation cannot affect the comparison results significantly.

*I hope that with the above revisions that address the major points raised above, the paper can become suitable for publication in ACP as the novelty of the work and the need for such data is high.*

We do appreciate the referee's comment about the value of long-term total OH reactivity measurements and we hope that removing the development of model-based correction factors to focus on the results satisfy the referee. Moreover, the revised manuscript has a much easier to read experimental part and it is an overall less cluttered paper. Further improvements in the figures should have improved further the readability of the manuscript.

**References**

Atkinson, R., Baulch, D. L., Cox, R. A., Hampson, R. F., Kerr, J. A. and Troe, J.: Evaluated Kinetic and Photochemical Data for Atmospheric Chemistry: Supplement IV. IUPAC Subcommittee on Gas Kinetic Data Evaluation for Atmospheric Chemistry, Journal of Physical and Chemical Reference Data, 21(6), 1125–1568, doi:10.1063/1.555918, 1992.

Boy, M., Mogensen, D., Smolander, S., Zhou, L., Nieminen, T., Paasonen, P., Plass-Dülmer, C., Sipilä, M., Petäjä, T., Mauldin, L., Berresheim, H. and Kulmala, M.: Oxidation of SO2 by stabilized Criegee intermediate (sCI) radicals as a crucial source for atmospheric sulfuric acid concentrations, Atmos. Chem. Phys., 13(7), 3865–3879, doi:10.5194/acp-13-3865-2013, 2013.

Fuchs, H., Novelli, A., Rolletter, M., Hofzumahaus, A., Pfannerstill, E. Y., Kessel, S., Edtbauer, A., Williams, J., Michoud, V., Dusanter, S., Locoge, N., Zannoni, N., Gros, V., Truong, F., Sarda-Esteve, R., Cryer, D. R., Brumby, C. A., Whalley, L. K., Stone, D., Seakins, P. W., Heard, D. E., Schoemaecker, C., Blocquet, M., Coudert, S., Batut, S., Fittschen, C., Thames, A. B., Brune, W. H., Ernest, C., Harder, H., Muller, J. B. A., Elste, T., Kubistin, D., Andres, S., Bohn, B., Hohaus, T., Holland, F., Li, X., Rohrer, F., Kiendler-Scharr, A., Tillmann, R., Wegener, R., Yu, Z., Zou, Q. and Wahner, A.: Comparison of OH reactivity measurements in the atmospheric simulation chamber SAPHIR, Atmos. Meas. Tech., 10(10), 4023–4053, doi:10.5194/amt-10-4023-2017, 2017.

Leino, K., Riuttanen, L., Nieminen, T., Maso, M. D., Väänänen, R., Pohja, T., Keronen, P., Järvi, L., Aalto, P. P., Virkkula, A., Kerminen, V.-M., Petäjä, T. and Kulmala, M.: Biomass-burning smoke episodes in Finland from eastern European wildfires, Boreal Env. Res., 19 (suppl. B), 275–292, 2014.

Michoud, V., Hansen, R. F., Locoge, N., Stevens, P. S. and Dusanter, S.: Detailed characterizations of the new Mines Douai comparative reactivity method instrument via laboratory experiments and modeling, Atmos. Meas. Tech., 8(8), 3537–3553, doi:10.5194/amt-8-3537-2015, 2015.

Nölscher, A. C., Sinha, V., Bockisch, S., Klüpfel, T. and Williams, J.: Total OH reactivity measurements using a new fast Gas Chromatographic Photo-Ionization Detector (GC-PID), Atmos. Meas. Tech., 5(12), 2981–2992, doi:10.5194/amt-5-2981-2012, 2012.

Praplan, A. P., Pfannerstill, E. Y., Williams, J. and Hellén, H.: OH reactivity of the urban air in Helsinki, Finland, during winter, Atmos. Env., 169, 150–161, doi:10.1016/j.atmosenv.2017.09.013, 2017.

Sinha, V., Williams, J., Crowley, J. N. and Lelieveld, J.: The Comparative Reactivity Method – a new tool to measure total OH Reactivity in ambient air, Atmos. Chem. Phys., 8(8), 2213–2227, doi:10.5194/acp-8-2213-2008, 2008.

Zhou, P., Ganzeveld, L., Taipale, D., Rannik, Ü., Rantala, P., Rissanen, M. P., Chen, D. and Boy, M.: Boreal forest BVOC exchange: emissions versus in-canopy sinks, Atmospheric Chemistry and Physics, 17(23), 14309–14332, doi:https://doi.org/10.5194/acp-17-14309-2017, 2017.

---

## Author Response (AR2)

**Authors' response**

We thank again the referee for his critical review of our work. We also thank the editor for coordinating the peer-review process. We provide here answers to the referee's and editor's comments. (Referee comments are in italics.)

**Anonymous referee #2**

*I acknowledge the efforts put in by the authors in revising the paper. However I still have very major concerns that have arisen with some of the revised plots made in the new version to address concerns raised in the original review. These concerns cast serious doubt on the quality and comparability of the OH reactivity measurements during the different seasons. I have listed these below:*
*1) Fig 2: Although the authors claim that the GC-PID dectector has no dependence on humidity, the calibration plots they provide in Fig 2 are not at all convincing. There are several major issues with it.*
*Looking at the right panel, one would surmise that the detector signal in June at 25 ppb of pyrrole was close to zero. The fits were arrived at using just three points and making the lines pass through zero, not even allowing for the fact there could be different residual noise in the detector under wet and dry conditions....the wet and dry calibration points were not even done at the same introduced pyrrole mixing ratios, which could have been helpful if the information were to be inferred using just 2-3 points to get a fit!*
*Coming to the left hand figure, it does have many more points but the lowest point even for 10 ppb pyrrole when the instrument was more sensitive in April at the start of their seasonal measurements for the long term deployment, shows a detector signal close to zero. The delta C3 (that is C3-C2) values would be typically less than 10 ppb for the ambient OH reactivity values measured during their study. How then can one distinguish the signal from noise beyond reasonable doubt and just trust the values blindly by applying statistics on the measured signals ?*

We regret the poor quality of Figure 2 in the revised manuscript. Following the referee's concern, we had a closer look at the underlying raw data and realized a mistake in the way the script that we used calculated pyrrole mixing ratios and corrected it (Fig. A1) for the second revision of the manuscript.

[Figure]

Figure A1. Left: GC-PID sensitivity for pyrrole used in the present study. Right: Same day test (30 June 2015) for GC-PID sensitivity to pyrrole under dry and humid conditions.

In the right panel, the wet and dry calibrations were indeed both derived introducing the same amount of pyrrole. The fits (with only 3 points) are now within 2.8 %, which corresponds to the measurement uncertainty with the GC-PID. In the left panel, the calibration of 30 July remains unchanged, while the calibration of 5 April was corrected. The slope of the fit is now 1797 a.u./ppbv (about 9% higher than the value in the revised manuscript). We revised all our calculations based on this revised sensitivity value in the second revision of the manuscript.

*2) The NO2 correction is still beyond my comprehension….first of all the plot in Fig3 shows change in C3 signal to be appreciable only starting at 20 ppb in the reactor. Considering there is large dilution of ambient air before it enters the reactor this would imply that much higher ambient NO2 of even 30 ppb would not change C3 by much. In this relatively clean forest site how much of the time period that the authors measured was characterized by 20-30 ppb NO2? How many data points? Even the reasoning they give for NO2 interference being important is not unclear. They assert NO2 photolysis produces NO which reacts with HO2 to give OH. But why woudn't the reaction of NO also occur with O3 which is more abundant in the reactor and competitive reaction wise, not give back NO2 and compensate this effect?*
*It does not seem as though they quantified reaction rates of both reactions for the range of conditions inside their reactor... I cannot understand why the authors wish to pursue with such experiments and analysis in the present work, and try to draw conclusion attributing corrections to a single factors despite the occurrence of several possible confounders during these experiments?*

This is a very legitimate concern from the referee. While the decrease of $C_3$ due to the introduction of $NO_2$ has been reported previously (Michoud et al., 2015; Praplan et al., 2017), we lost sight of its relevance to the present study. The referee is right to ask how much of the data points were corrected for this and after verification it appears to be only less than 4%. For this reason, we have removed this correction from the data analysis. We accordingly removed its description and any associated discussion and updated all the results in the second revision of the manuscript.

*I am regret very much that this version is still lacking as this is a very good research group which is well known for its VOC measurement expertise in the world. Just this work in my opinion has serious shortcomings for me to have confidence in the results and conclusions of the study.*

We hope that this second revision addresses adequately the referee's main concerns and we appreciate founded criticism, regardless of the reputation of our research group.

**Editor**

*On the revised manuscript, Reviewer #2 shows a big concern, particularly about Figure 2 (small number of data and points largely deviated from linearity) and thus the quantitativeness of the*

*determined reactivity from the measurements. The present editor also suspects that the measurements suffer from large uncertainties (not well specified), affecting solidity of the conclusion.*

We regret that Fig. 2 of the revised manuscript was of such poor quality, due to a mistake in a script used in our data analysis. We have redone the figure and do hope that correcting our mistake answers the referee's concerns (as described above). We also detail the uncertainties better in the main text and with additional plots in the Appendix of the second revision of the manuscript.

*The authors also need to address the issue that the present editor originally requested at the timing of resubmission, to verify if the concentrations of secondary organic species are adequately modeled. Independent measurements of aldehydes or peroxides should be used to test the adequacy of the model. Otherwise the modeled levels of the secondary species could become very arbitrary and the conclusion about the presence of missing reactivity is weak.*

Our options to validate the model in the way that the editor suggests are very limited, which is why we refered to previous work by (Zhou et al., 2017a, 2017b) regarding the validation of the model. The main reason is the lack of measurements for specific secondary compounds during the measurement period of this study. No PTR-MS was operating at the SMEAR II station during that period. This is now mentioned in the main text of our manuscript. Also no measurements of any peroxide or specific aldehydes (such as pinonaldehyde, for instance) are available during that time either. Comparing non-specific oxidation products might be misleading as not all the precursors are included in the model and these compounds could have been transported. Nevertheless, nopinone, a reaction production of β-pinene was measured by GC-MS. To provide the reader a verification of the secondary organic species produced in the model, we have added a plot in the Appendix to compare measured and modelled nopinone (Fig. A2) and added information related to earlier model validations for organic species with SOSAA at SMEAR II. As it is the most we can do with the measurements available, we hope that this and the previous studies suffice to convince the editor that the modelled secondary species agree reasonably well with measurements.

[Figure]

Figure A2. Mixing ratio of nopinone measured (red line and shaded area for 50% uncertainty) and modelled (blue line) for the three modelled periods in April, May, and July (from left to right).

We modified the text at the end of section 2.6 which now reads: "The latter describes the explicit simulation of the loss of every compound in the model by dry deposition inside the canopy for all height levels and provides a detailed comparison of measured and modelled fluxes of certain selected VOCs including some secondary organic species at SMEAR II."

In addition we added the following text to improve the third paragraph of section 3.4: "To check the model performance for the concentrations of the secondary organic species, we compared nopinone measured by GC-MS with the model output. The plots for all three selected periods (Fig. C1 in the Appendix) show that the trend and of the model concentrations for nopinone are comparable to the measured values and the absolute values are mostly inside estimated 50% uncertainties of the

measurements.
The only exceptions are the difference during daytimes in May where the measurements show a very strong decrease in the morning but the model follows these behavior much weaker. No other specific secondary species is available for such comparison due to the lack of measurements."

*Only when the authors could provide appropriate reply to the points above and other important points that the reviewer raised, further consideration will be made.*

We hope that we could reply appropriately to the points raised in the second revision of our manuscript.

**References**

[revised manuscript text omitted]

**Appendix C:  **Comparison of measured and modelled nopinone**

[Figure]

**Figure C1.** Mixing ratio of nopinone measured (red line and shaded area for 50 % uncertainty) and modelled (blue line) for the three modelled periods in April, May, and July (from left to right).

**Appendix D:** **Details for the modelled periods**

Table D1: Averages of individual compounds mixing ratios [ppt$_v$] and calculated OH reactivity, $R_{OH}$ [s$^{-1}$], and group $R_{OH}$ for the t[...] studied with SOSAA. 'n.d.' means 'not detected' and 'n.m.' means 'not measured'.

| Compound | Mix. ratio [ppt$_v$] mean | (std) | 22 – 25 May Mix. ratio [ppt$_v$] mean | (std) | 22 – 25 May $R_{OH}$ [s$^{-1}$] mean | (std) | 29 – 30 April Mix. ratio [ppt$_v$] mean | (std) | 29 – 30 April $R_{OH}$ [s$^{-1}$] mean | (std) |
|---|---|---|---|---|---|---|---|---|---|---|
| *Alkanes* | | | | | 0.00035 | (0.00024) | | | 0.065 | (0.013) |
| ethane | n.m. | (-) | n.m. | (-) | - | (-) | 2775 | (78) | 0.0115 | (0.0003) |
| propane | n.m. | (-) | n.m. | (-) | - | (-) | 576 | (79) | 0.014 | (0.002) |
| n-butane | n.m. | (-) | n.m. | (-) | - | (-) | 139 | (48) | 0.0078 | (0.0026) |
| 2-methylpropane | n.m. | (-) | n.m. | (-) | - | (-) | 88 | (31) | 0.0045 | (0.0016) |
| n-pentane | n.m. | (-) | n.m. | (-) | - | (-) | 61 | (13) | 0.0055 | (0.0012) |
| 2-methylbutane | n.m. | (-) | n.m. | (-) | - | (-) | 112 | (16) | 0.011 | (0.002) |
| n-hexane | n.m. | (-) | n.m. | (-) | - | (-) | 23 | (7) | 0.0028 | (0.0009) |
| 2-methylpentane | n.m. | (-) | n.m. | (-) | - | (-) | 25 | (7) | 0.0034 | (0.0010) |
| n-heptane | 0.25 | (0.21) | 0.54 | (0.35) | 0.000058 | (0.000109) | 5.6 | (1.9) | 0.0020 | (0.0007) |
| n-octane | 1.9 | (0.4) | 1.0 | (0.4) | 0.00019 | (0.00009) | 7.3 | (2.5) | 0.0015 | (0.0005) |
| n-nonane | 0.13 | (0.07) | 0.43 | (0.14) | 0.00010 | (0.00004) | 3.3 | (1.2) | 0.00081 | (0.00030) |
| n-decane | n.d. | (-) | n.d. | (-) | - | (-) | 2.3 | (1.1) | 0.00064 | (0.000031) |
| *Alkenes* | | | | | - | (-) | | | 0.38 | (0.04) |
| ethene | n.m. | (-) | n.m. | (-) | - | (-) | 354 | (26) | 0.077 | (0.006) |
| propene | n.m. | (-) | n.m. | (-) | - | (-) | 135 | (6) | 0.11 | (0.01) |
| 1-butene | n.m. | (-) | n.m. | (-) | - | (-) | 47 | (4) | 0.042 | (0.004) |
| *trans*-2-butene | n.m. | (-) | n.m. | (-) | - | (-) | 46 | (8) | 0.086 | (0.016) |
| *cis*-2-butene | n.m. | (-) | n.m. | (-) | - | (-) | 27 | (4) | 0.043 | (0.007) |
| 1,3-butadiene | n.m. | (-) | n.m. | (-) | - | (-) | n.d. | (-) | - | (-) |
| 1-pentene | n.m. | (-) | n.m. | (-) | - | (-) | 35 | (7) | 0.025 | (0.005) |
| *trans*-2-pentene | n.m. | (-) | n.m. | (-) | - | (-) | n.d. | (-) | - | (-) |
| ethyne | n.m. | (-) | n.m. | (-) | - | (-) | 260 | (21) | 0.0051 | (0.0004) |
| *Aromatics* | | | | | 0.012 | (0.009) | | | 0.038 | (0.014) |
| benzene | 15 | (4) | 12 | (3) | 0.00035 | (0.00010) | 93 | (16) | 0.0028 | (0.0005) |
| toluene | 22 | (6) | 32 | (10) | 0.0046 | (0.0017) | 37 | (9) | 0.0058 | (0.0014) |
| ethylbenzene | 6.5 | (1.7) | 2.7 | (0.8) | 0.00047 | (0.00015) | 10 | (2) | 0.0018 | (0.0004) |
| p/m-xylene | 11 | (2) | 3.1 | (2.2) | 0.0014 | (0.0011) | 14 | (7) | 0.0067 | (0.0033) |
| o-xylene | 2.3 | (1.0) | 1.0 | (0.9) | 0.00017 | (0.00029) | 5.8 | (1.9) | 0.0020 | (0.0007) |
| styrene | 8.8 | (8.2) | 1.8 | (1.4) | 0.0025 | (0.0020) | 7.6 | (3.2) | 0.011 | (0.005) |
| 2-ethyltoluene | 0.61 | (0.23) | 0.42 | (0.23) | 0.000053 | (0.000077) | 1.2 | (0.3) | 0.00036 | (0.00010) |
| 3-ethyltoluene | 0.28 | (0.16) | 0.80 | (1.53) | 0.00031 | (0.0065) | n.d. | (-) | - | (-) |
| 4-ethyltoluene | 0.35 | (0.29) | 0.27 | (0.12) | 0.000032 | (0.000045) | 0.12 | (0.05) | 0.000036 | (0.000016) |

| | 29 – 30 April | | | | 22 – 25 May | | | |
|---|---|---|---|---|---|---|---|---|
| | Mixing ratio [pptv] | | $R_{OH}$ [s$^{-1}$] | | $R_{OH}$ [s$^{-1}$] | | Mixing ratio [pptv] | |
| | mean | (std) | mean | (std) | mean | (std) | mean | (std) |
| 1,2,3-trimethylbenzene | 2.2 | (0.8) | 0.0019 | (0.0007) | 0.0019 | (0.0020) | 1.4 | (0.5) |
| 1,2,4-trimethylbenzene | 3.3 | (1.0) | 0.0028 | (0.0009 0.0008) | 0.00028 | (0.00040) | 0.44 | (0.21) |
| 1,3,5-trimethylbenzene | 1.3 | (0.7) | 0.0021 | (0.0011) | 0.00018 | (0.00031) | 0.24 | (0.10) |
| isoprene | 3.9 | (3.0) | 0.010 | (0.009) | 0.020 | (0.017) | 29 | (14) |
| *Monoterpenoids* | | | *0.53* | *(0.39)* | *0.47* | *(0.53)* | | |
| α-pinene | 223 221 | (142 143) | 0.33 | (0.22) | 0.16 | (0.19) | 635 | (318) |
| β-pinene | 28 27 | (26) | 0.064 0.059 | (0.058) | 0.047 | (0.053) | 105 | (74) |
| camphene | 22 | (16) | 0.030 | (0.022) | 0.037 | (0.038) | 52 | (33) |
| Δ$^3$-carene | 44 | (35) | 0.10 | (0.08) | 0.16 | (0.18) | 224 | (141) |
| p-cymene | 5.5 | (2.3) | 0.0021 | (0.0009) | 0.0080 | (0.0087) | 11 | (5) |
| limonene | 1.8 | (1.4) | 0.0037 0.0038 | (0.0061) | 0.050 | (0.058) | 97 | (65) |
| terpinolene | n.d. | (-) | - | - | 0.00068 | (0.00160) | 11 | (7) |
| myrcene | 0.25 | (0.26) | 1.6e-12 | (1.9e-12) | 1.8e-11 | (2.2e-11) | 14 | (8) |
| 1,8-cineol | 2.6 | (2.4) | 0.00076 | (0.00069) | 0.0033 | (0.0024) | 22 | (9) |
| bornylacetate | 0.31 | (0.20) | 0.00011 | (0.00007) | 0.00033 | (0.00038) | 2.7 | (1.3) |
| *Sesquiterpenes* | | | *0.0015* | *(0.0030)* | *0.022* | *(0.024)* | | |
| longicyclene | 0.32 | (0.27) | 0.000079 0.000078 | (0.000066) | 0.00009 0.000089 | (0.00010 0.000104) | 0.78 | (0.37) |
| iso-longifolene | 0.0600 | (0.0003) | 0.000042 | (0.000068) | 0.000036 | (0.000168) | n.d. | (-) |
| β-farnesene | n.d. | (-) | - | - | - | (-) | 4.0 | (1.4) |
| β-caryophyllene | 0.94 | (0.61 0.60) | 0.0013 | (0.0027) | 0.020 | (0.023) | 28 | (16) |
| α-humulene | 0.0514 | (0.0001) | 0.000071 | (0.000149) | 0.0014 | (0.0011) | n.d. | (-) |
| SQT1* | n.d. | (-) | - | - | - | - | 2.7 | (1.5) |
| SQT2* | n.d. | (-) | - | - | - | - | 5.4 | (3.3) |
| SQT3* | n.d. | (-) | - | - | - | - | 4.5 | (2.5) |
| SQT4* | n.d. | (-) | - | - | - | - | 12 | (6) |
| *GLVs* | | | *-* | *(-)* | *0.0021* | *(0.0020)* | | |
| 1-hexanol | n.d. | (-) | - | - | - | - | 7.8 | (4.2) |
| cis-2-hexen-1-ol | n.d. | (-) | - | - | - | - | n.d. | (-) |
| trans-2-hexen-1-ol | n.d. | (-) | - | - | - | - | n.d. | (-) |
| cis-3-hexen-1-ol | n.d. | (-) | - | - | - | - | 4.3 | (2.4) |
| trans-3-hexen-1-ol | n.d. | (-) | - | - | - | - | 6.7 | (1.5) |
| trans-2-hexenal | n.d. | (-) | - | - | 0.0021 | (0.0020) | 3.3 | (2.1) |
| hexylacetate | n.d. | (-) | - | - | - | - | n.d. | (-) |
| cis-3-hexenylacetate | n.d. | (-) | - | - | - | - | n.d. | (-) |
| n.d. | (-) | - | - | (-) | - | - | | |
| trans-2-hexenyl-acetate | n.d. | (-) | - | - | - | - | n.d. | (-) |

| | 29 – 30 April | | | | 22 – 25 May | | | | (period label cut off) | |
|---|---|---|---|---|---|---|---|---|---|---|
| | Mixing ratio [ppt$_v$] | | $R_{OH}$ [s$^{-1}$] | | Mixing ratio [ppt$_v$] | | $R_{OH}$ [s$^{-1}$] | | Mixing ratio [ppt$_v$] | |
| | mean | (std) | mean | (std) | mean | (std) | mean | (std) | mean | (std) |
| *Aldehydes* | | | *0.10* | *(0.07)* | | | *0.075* | *(0.051)* | | |
| formaldehyde | 122 | (111) | 0.028 | (0.025) | n.m. | (-) | - | (-) | 620 | (90) |
| acetaldehyde | 16.5 | (0.1) | 0.0018 | (0.0030) | n.m. | (-) | - | (-) | 342 | (62) |
| propanal | 86 | (32) | 0.046 | (0.017) | 93 | (49) | 0.040 | (0.027) | 112 | (36) |
| butanal | n.d. | (-) | - | - | 4.7 | (1.5) | 0.00039 | (0.00104) | 17 | (26) |
| pentanal | 19 | (6) | 0.015 | (0.005) | 24 | (20) | 0.011 | (0.015) | 41 | (16) |
| hexanal | 8.03 | (0.04) | 0.0017 | (0.0029) | 7.3 | (3.3) | 0.0052 | (0.0025) | 17 | (8) |
| heptanal | n.d. | (-) | - | - | 5.9 | (1.5) | 0.0043 | (0.0013) | 0.10 | (0.08) |
| octanal | n.d. | (-) | - | - | 4.2 | (1.0) | 0.0032 | (0.0009) | 6.1 | (1.7) |
| nonanal | n.d. | (-) | - | - | 2.8 | (1.0) | 0.0024 | (0.0009) | 9.4 | (4.1) |
| decanal | n.d. | (-) | - | - | 3.1 | (0.8) | 0.0026 | (0.0008) | 9.9 | (3.0) |
| methacrolein | 8.0 | (1.4) | 0.0030 | (0.0033) | 8.0 | (3.3) | 0.0058 | (0.0025) | 7.1 | (6.5) |
| crotonaldehyde | 1.6 | (0.1) | 0.00079 | (0.00071) | n.m. | (-) | - | (-) | n.d. | (-) |
| benzaldehyde | 26 | (2) | 0.0016 | (0.0035) | n.m. | (-) | - | (-) | n.d. | (-) |
| toluadehyde | 75 | (7) | 0.0060 | (0.0129) | n.m. | (-) | - | (-) | n.d. | (-) |
| *Alcohols* | | | *0.086* | *(0.080)* | | | *0.21* | *(0.56)* | | |
| isopropanol | 26 | (6) | 0.0035 | (0.0008) | 37 | (29) | 0.0041 | (0.0039) | 171 | (95) |
| 1-butanol | 366 | (349) | 0.083 | (0.079) | 1122 | (2704) | 0.21 | (0.55) | 614 | (745) |
| 1-pentanol | n.d. | (-) | - | - | 3.7 | (1.4) | 0.000065 | (0.000285) | 8.6 | (3.3) |
| 1-penten-3-ol | n.d. | (-) | - | - | 1.9 | (0.7) | 0.00041 | (0.00120) | 3.6 | (2.3) |
| 3-methyl-2-buten-1-ol | n.d. | (-) | - | - | n.d. | (-) | - | (-) | n.d. | (-) |
| 1-octen-3-ol | n.d. | (-) | - | - | n.d. | (-) | - | (-) | 1.5 | (0.4) |
| 2-methyl-3-buten-2-ol (MBO) | 5.4 | (4.6) | 0.021 | (0.018) | 15 | (16) | 0.054 | (0.058) | 47 | (28) |
| *Other carbonyls* | | | *0.014* | *(0.018)* | | | *0.0012* | *(0.0019)* | | |
| acetone (and acrolein) | 3060 | (4141) | 0.012 | (0.017) | n.m. | (-) | - | (-) | 9161 | (1632) |
| 6-methyl-2-hepten-3-one | n.d. | (-) | - | (-) | n.d. | (-) | - | (-) | 1.5 | (0.6) |
| methyl ethyl ketone (MEK) | n.d. | (-) | - | (-) | n.d. | (-) | - | (-) | 9.0 | (0.3) |
| butylacetate | 2.9 | (1.3) | 0.000051 | (0.000143) | n.d. | (-) | - | (-) | n.d. | (-) |
| 4-acetyl-1-methylcyclohexene | n.d. | (-) | - | - | 1.3 | (0.6) | 0.00022 | (0.00105) | 4.5 | (4.0) |
| nopinone | 4.8 | (3.2) | 0.0018 | (0.0012) | 2.9 | (2.5) | 0.0010 | (0.0009) | 10 | (4) |
| *Organic acids* | | | *0.071* | *(0.013)* | | | *0.024* | *(0.018)* | | |
| acetic acid | 2800 | (446) | 0.060 | (0.008) | 1507 | (430) | 0.020 | (0.014) | 3007 | (283) |
| propanoic acid | 142 | (25) | 0.0044 | (0.0008) | 84 | (26) | 0.0018 | (0.0013) | 160 | (52) |
| butanoic acid | 98 | (37) | 0.0046 | (0.0017) | 58 | (33) | 0.0019 | (0.0017) | 152 | (26) |
| isobutanoic acid | n.d. | (-) | - | - | n.d. | (-) | - | - | 32 | (14) |
| pentanoic acid | 20 | (13) | 0.0016 | (0.0015) | 21 | (11) | 0.00055 | (0.00109) | 176 | (36) |

| | 29 – 30 April Mixing ratio [ppt$_v$] mean | (std) | $R_{OH}$ [s$^{-1}$] mean | (std) | 22 – 25 May $R_{OH}$ [s$^{-1}$] mean | (std) | Mixing ratio [ppt$_v$] mean | (std) |
|---|---|---|---|---|---|---|---|---|
| isopentanoic acid | 1.0 | (0.2) |  0.0000013 | (0.0000122) | 0.0000025 | (0.0000234) | 8.5 | (5.2) |
|  hexanoic acid |  5.9 | () | −0.000052 | (−0.000216) | −0.000055 | (−0.000275) |  35 | () |
|  4-methylpentanoic acid |  n.d. | ( -) |  - | ( -) |  - | ( -) |  n.d. | ( -) |
| heptanoic acid | n.d. | (-) | - | (-) | - | (-) | 13 | (3) |
| *Inorganics* | | | *1.2* | *(0.2)* | *1.0* | *(0.2)* | | |
| NO | 77 | (41) |  0.012 | (0.012) | 0.013 | (0.013) | 69 | ( 43) |
| NO$_2$ |  449 | ( 374) | 0.14 | (0.12) | 0.11 | (0.08) | 149 | (94) |
| O$_3$ |  4.3e4 | ( 9e3) | 0.066 | (0.015) | 0.69 | (0.014) | 2.7e4 | (8.9e3) |
| SO$_2$ | 53 | (60) |  0.00090 | ( 0.00131) | 0.00058 | (0.00054) | 74 | (114) |
| CO | 1.29e5 | ( 9e3) | 0.72 | (0.06) | 0.58 | (0.03) |  1.4e5 | ( 2e4) |
| CH$_4$ |  1.938e6 | ( 2e3) | 0.23 | (0.01) | 0.26 | (0.02) |  1.9e6 | ( 2.2e-16) |
| *Model OVOCs* | | | * 0.19* | *(0.06)* | * 0.20* | *( 0.09)* | | |
| *Model inorganics* | | | *0.057* | *( 0.008)* | * 0.084* | *( 0.026)* | | |
| *Missing* | | | * 61* | *( 40)* | * 0.13* | *(78)* | | |
| *Total* | | | * 9.64* | *( 341)* | * 15* | *( 10)* | | |

655 * quantified with $\beta$-caryophyllene calibration and an estimated reaction coefficient (1e-10 cm$^3$ s$^{-1}$)

---

## Author Response (AR3)

**Authors' response**

We thank the co-editor for his examination of the revised manuscript and are glad that we were able to address the previous critical points. Regarding the necessary minor revision, our answers can be found below. (Co-editor comments are in italics.)

*1. Figures 6 and 8 are opposite?*

Yes, these two Figures got mixed up by accident. It is now fixed.

*2. I understand that the measured OH reactivities are shifted, as the sensitivity changed. But why ambient conditions such as [MT], [SQT], and even precipitation and RH in Table 1 changed from previous version? (same for Table D1)*

The averages for ambient conditions are calculated for periods when OH reactivity is above the detection limit. Due to the changes in OH reactivity values, the periods when the OH reactivity is above the detection limit (overlap) has changed slightly, leading to small changes in the average values for ambient conditions. For most small values, though, the change is not visible due to the rounding of the values.

*3. Table D1. Better to use 3.5x10^-4 instead of 0.00035. All same for small values.*

In table D1, we use now the scientific notation for all average values smaller than 0.001. (We kept the same notation for standard deviation values smaller than 0.001 if the average value is larger than that this threshold.)

[revised manuscript text omitted]
 | 5.6 | (1.9) | 0.0020 | (0.0007) | 0.54 | (0.35) | 0.00005 5.8e-5 | (0.00004 10.9e-5) | 0.25 | (0.21) | 0.00004 1.1e-4 |
| *n*-octane | 7.3 | (2.5) | 0.0015 | (0.0005) | 1.0 | (0.4) | 0.00009 1.9e-4 | (0.00009 0.9e-4) | 1.9 | (0.4) | 0.00003 3.2e-4 |
| *n*-nonane | 3.3 | (1.2) | 0.0008+8.1e-4 | (0.00030 3.0e-4) | 0.43 | (0.14) | 0.00004 1.0e-4 | (0.00004 0.4e-4) | 0.13 | (0.07) | 0.00002 2.4e-5 |
| *n*-decane | 2.3 | (1.1) | 0.0006+6.4e-4 | (0.00034 3.1e-4) | n.d. | (-) | - | (-) | n.d. | (-) | |
| *Alkenes* | | | 0.38 | (0.04) | | | - | (-) | | | 0.02 |
| ethene | 354 | (26) | 0.077 | (0.006) | n.m. | (-) | - | (-) | n.m. | (-) | |
| propene | 135 | (6) | 0.11 | (0.01) | n.m. | (-) | - | (-) | n.m. | (-) | |
| 1-butene | 47 | (4) | 0.042 | (0.004) | n.m. | (-) | - | (-) | n.m. | (-) | |
| *trans*-2-butene | 46 | (8) | 0.086 | (0.016) | n.m. | (-) | - | (-) | n.m. | (-) | |
| *cis*-2-butene | 27 | (4) | 0.043 | (0.007) | n.m. | (-) | - | (-) | n.m. | (-) | |
| 1,3-butadiene | n.d. | (-) | - | (-) | n.m. | (-) | - | (-) | n.m. | (-) | |
| 1-pentene | 35 | (7) | 0.025 | (0.005) | n.m. | (-) | - | (-) | n.m. | (-) | |
| *trans*-2-pentene | n.d. | (-) | - | (-) | n.m. | (-) | - | (-) | n.m. | (-) | |
| ethyne | 260 | (21) | 0.0051 | (0.0004) | n.m. | (-) | - | (-) | n.m. | (-) | |
| *Aromatics* | | | 0.038 | (0.014) | | | 0.012 | (0.009) | | | 0.02 |
| benzene | 93 | (16) | 0.0028 | (0.0005) | 12 | (3) | 0.00035 3.5e-4 | (0.00001 1.0e-4) | 15 | (4) | 0.00036 3.6e-4 |
| toluene | 37 | (9) | 0.0058 | (0.0014) | 32 | (10) | 0.0046 | (0.0017) | 22 | (6) | 0.0002 |
| ethylbenzene | 10 | (2) | 0.0018 | (0.0004) | 2.7 | (0.8) | 0.00047 4.7e-4 | (0.00001 5.1e-4) | 6.5 | (1.7) | 0.00093 9.3e-4 |
| *p/m*-xylene | 14 | (7) | 0.0067 | (0.0033) | 3.1 | (2.2) | 0.0014 | (0.0011) | 11 | (2) | 0.004 |
| *o*-xylene | 5.8 | (1.9) | 0.0020 | (0.0007) | 1.0 | (0.9) | 0.00017 1.7e-4 | (0.00002 2.9e-4) | 2.3 | (1.0) | 0.00063 6.3e-4 |
| styrene | 7.6 | (3.2) | 0.011 | (0.005) | 1.8 | (1.4) | 0.0025 | (0.0020) | 8.8 | (8.2) | 0.01 |
| 2-ethyltoluene | 1.2 | (0.3) | 0.00036 3.6e-4 | (0.00001 1.0e-4) | 0.42 | (0.23) | 0.00005 5.3e-5 | (0.00007 7.7e-5) | 0.61 | (0.23) | 0.00015 1.5e-4 |
| 3-ethyltoluene | n.d. | (-) | - | (-) | 0.80 | (1.53) | 0.00004 3.1e-4 | (0.00005 6.5e-4) | 0.28 | (0.16) | 0.00011 1.1e-4 |
| 4-ethyltoluene | 0.12 | (0.05) | 0.00036 3.6e-5 | (0.00016 1.6e-4) | 0.27 | (0.12) | 0.00003 3.2e-5 | (0.00005 4.5e-5) | 0.35 | (0.29) | 0.00032 3.2e-5 |

| | 29 – 30 April | | | | 22 – 25 May | | | | 24 – 26 July | | |
|---|---|---|---|---|---|---|---|---|---|---|---|
| | Mixing ratio [pptv] | | $R_{OH}$ [s$^{-1}$] | | Mixing ratio [pptv] | | $R_{OH}$ [s$^{-1}$] | | Mixing ratio [pptv] | | $R_{OH}$ [s$^{-1}$] |
| | mean | (std) | mean | (std) | mean | (std) | mean | (std) | mean | (std) | mean |
| 1,2,3-trimethylbenzene | 2.2 | (0.8) | 0.0019 | (0.0007) | 2.4 | (2.4) | 0.0019 | (0.0020) | 1.4 | (0.5) | 0.00094 |
| 1,2,4-trimethylbenzene | 3.3 | (1.0) | 0.0028 | (0.0008) | 0.56 | (0.52) | 0.00028 | (2.8e-4) | 0.44 | (0.21) | 0.00029 |
| 1,3,5-trimethylbenzene | 1.3 | (0.7) | 0.0021 | (0.0011) | 0.37 | (0.21) | 0.00018 | (1.8e-4) | 0.24 | (0.10) | 0.00024 |
| isoprene | 3.9 | (3.0) | 0.010 | (0.009) | 8.0 | (6.4) | 0.020 | (0.017) | 29 | (14) | 0.066 |
| *Monoterpenoids* | | | *0.53* | *(0.39)* | | | *0.47* | *(0.53)* | | | *1.* |
| α-pinene | 221 | (143) | 0.33 | (0.22) | 120 | (134) | 0.16 | (0.19) | 635 | (318) | 0.7 |
| β-pinene | 27 | (26) | 0.059 | (0.058) | 24 | (27) | 0.047 | (0.053) | 105 | (74) | 0.1 |
| camphene | 22 | (16) | 0.030 | (0.022) | 29 | (29) | 0.037 | (0.038) | 52 | (33) | 0.05 |
| Δ³-carene | 44 | (35) | 0.10 | (0.08) | 72 | (82) | 0.16 | (0.18) | 224 | (141) | 0.4 |
| p-cymene | 5.5 | (2.3) | 0.0021 | (0.0009) | 23 | (24) | 0.0080 | (0.0087) | 11 | (5) | 0.003 |
| limonene | 1.8 | (1.4) | 0.0038 | (0.0061) | 12 | (14) | 0.050 | (0.058) | 97 | (65) | 0.3 |
| terpinolene | n.d. | (-) | - | | 0.53 | (0.37) | 0.00069 | (6.0e-4) | 11 | (7) | 0.05 |
| myrcene | 0.25 | (0.26) | 1.6e-12 | (1.9e-12) | 4.2 | (3.0) | 1.8e-11 | (2.2e-11) | 14 | (8) | 8.0e-1 |
| 1,8-cineol | 2.6 | (2.4) | 0.00076 | (6.9e-4) | 12 | (9) | 0.0033 | (0.0024) | 22 | (9) | 0.0005 |
| bornylacetate | 0.31 | (0.20) | 0.00011 | (1.1e-4) | 1.7 | (0.9) | 0.00033 | (3.8e-4) | 2.7 | (1.3) | 0.00075 |
| *Sequiterpenes* | | | *0.0015* | *(0.0030)* | | | *0.022* | *(0.024)* | | | *0.1* |
| longicyclene | 0.32 | (0.27) | 0.000078 | (7.8e-5) | 0.81 | (0.27) | 0.000089 | (10.4e-5) | 0.78 | (0.37) | 0.00015 |
| iso-longifolene | 0.0600 | (0.0003) | 0.000042 | (4.2e-5) | 0.28 | (0.13) | 0.000036 | (16.8e-5) | n.d. | (-) | |
| β-farnesene | n.d. | (-) | - | | n.d. | (-) | - | | 4.0 | (1.4) | 0.01 |
| β-caryophyllene | 0.94 | (0.60) | 0.0013 | (0.0027) | 7.3 | (3.7) | 0.020 | (0.023) | 28 | (16) | 0.1 |
| α-humulene | 0.0514 | (0.0001) | 0.000071 | (7.1e-5) | 0.21 | (0.15) | 0.0014 | (0.0011) | n.d. | (-) | |
| SQT1* | n.d. | (-) | - | | n.d. | (-) | - | | 2.7 | (1.5) | 0.005 |
| SQT2* | n.d. | (-) | - | | n.d. | (-) | - | | 5.4 | (3.3) | 0.01 |
| SQT3* | n.d. | (-) | - | | n.d. | (-) | - | | 4.5 | (2.5) | 0.0007 |
| SQT4* | n.d. | (-) | - | | n.d. | (-) | - | | 12 | (6) | 0.01 |
| *GLVs* | - | | - | | | | *0.0021* | *(0.0020)* | | | *0.01* |
| 1-hexanol | n.d. | (-) | - | | n.d. | (-) | - | | 7.8 | (4.2) | 0.001 |
| cis-2-hexen-1-ol | n.d. | (-) | - | | n.d. | (-) | - | | n.d. | (-) | |
| trans-2-hexen-1-ol | n.d. | (-) | - | | n.d. | (-) | - | | n.d. | (-) | |
| cis-3-hexen-1-ol | n.d. | (-) | - | | n.d. | (-) | - | | 4.3 | (2.4) | 0.0002 |
| trans-3-hexen-1-ol | n.d. | (-) | - | | n.d. | (-) | - | | 6.7 | (1.5) | 0.0006 |
| trans-2-hexenal | n.d. | (-) | - | | 2.4 | (1.8) | 0.0021 | (0.0020) | 3.3 | (2.1) | 0.002 |
| hexylacetate | n.d. | (-) | - | | n.d. | (-) | - | | n.d. | (-) | |
| cis-3-hexenylacetate | n.d. | (-) | - | | n.d. | (-) | - | | | (-) | |
| n.d. | (-) | - | - | | | | | | | | |
| trans-2-hexenyl-acetate | n.d. | (-) | - | | n.d. | (-) | - | | n.d. | (-) | |

| | 29 – 30 April | | | | 22 – 25 May | | | | 24 – 26 July | | |
|---|---|---|---|---|---|---|---|---|---|---|---|
| | Mixing ratio [ppt_v] | | $R_{OH}$ [s$^{-1}$] | | Mixing ratio [ppt_v] | | $R_{OH}$ [s$^{-1}$] | | Mixing ratio [ppt_v] | | $R_{OH}$ [s$^{-1}$] |
| | mean | (std) | mean | (std) | mean | (std) | mean | (std) | mean | (std) | mean |
| *Aldehydes* | | | *0.10* | *(0.07)* | | | *0.075* | *(0.051)* | | | *0.3* |
| formaldehyde | 122 | (111) | 0.028 | (0.025) | n.m. | (-) | - | - | 620 | (90) | 0.1 |
| acetaldehyde | 16.5 | (0.1) | 0.0018 | (0.0030) | n.m. | (-) | - | - | 342 | (62) | 0.1 |
| propanal | 86 | (32) | 0.046 | (0.017) | 93 | (49) | 0.040 | (0.027) | 112 | (36) | 0.01 |
| butanal | n.d. | (-) | - | (-) | 4.7 | (1.5) | 3.9e-4 | (1.0e-4) | 17 | (26) | 0.008 |
| pentanal | 19 | (6) | 0.015 | (0.005) | 24 | (20) | 0.011 | (0.015) | 41 | (16) | 0.02 |
| hexanal | 8.03 | (0.04) | 0.0017 | (0.0029) | 7.3 | (3.3) | 0.0052 | (0.0025) | 17 | (8) | 0.01 |
| heptanal | n.d. | (-) | - | - | 5.9 | (1.5) | 0.0043 | (0.0013) | 0.10 | (0.08) | 1.7e-6 |
| octanal | n.d. | (-) | - | - | 4.2 | (1.0) | 0.0032 | (0.0009) | 6.1 | (1.7) | 0.004 |
| nonanal | n.d. | (-) | - | - | 2.8 | (1.0) | 0.0024 | (0.0009) | 9.4 | (4.1) | 0.006 |
| decanal | n.d. | (-) | - | - | 3.1 | (0.8) | 0.0026 | (0.0008) | 9.9 | (3.0) | 0.007 |
| methacrolein | 8.0 | (1.4) | 0.0030 | (0.0033) | 8.0 | (3.3) | 0.0058 | (0.0025) | 7.1 | (6.5) | 0.004 |
| crotonaldehyde | 1.6 | (0.1) | 7.9e-4 | (7.1e-4) | n.m. | (-) | - | (-) | n.d. | (-) | |
| benzaldehyde | 26 | (2) | 0.0016 | (0.0035) | n.m. | (-) | - | - | n.d. | (-) | |
| tolualdehyde | 75 | (7) | 0.0060 | (0.0129) | n.m. | (-) | - | (-) | n.d. | (-) | |
| *Alcohols* | | | *0.086* | *(0.080)* | | | *0.21* | *(0.56)* | | | *0.05* |
| isopropanol | 26 | (6) | 0.0035 | (0.0008) | 37 | (29) | 0.0041 | (0.0039) | 171 | (95) | 0.006 |
| 1-butanol | 366 | (349) | 0.083 | (0.079) | 1122 | (2704) | 0.21 | (0.55) | 614 | (745) | 0.04 |
| 1-pentanol | n.d. | (-) | - | (-) | 3.7 | (1.4) | 6.5e-5 | (2.85e-5) | 8.6 | (3.3) | 7.6e-4 |
| 1-penten-3-ol | n.d. | (-) | - | (-) | 1.9 | (0.7) | 4.1e-4 | (1.2e-4) | 3.6 | (2.3) | 0.001 |
| 3-methyl-2-buten-1-ol | n.d. | (-) | - | (-) | n.d. | (-) | - | (-) | n.d. | (-) | |
| 1-octen-3-ol | n.d. | (-) | - | (-) | n.d. | (-) | - | (-) | 1.5 | (0.4) | |
| 2-methyl-3-buten-2-ol (MBO) | 5.4 | (4.6) | 0.021 | (0.018) | 15 | (16) | 0.054 | (0.058) | 47 | (28) | 0.1 |
| *Other carbonyls* | | | *0.014* | *(0.018)* | | | *0.0012* | *(0.0019)* | | | *0.05* |
| acetone (and acrolein) | 3060 | (4141) | 0.012 | (0.017) | n.m. | (-) | - | (-) | 9161 | (1632) | 0.03 |
| 6-methyl-2-hepten-3-one | n.d. | (-) | - | (-) | n.d. | (-) | - | (-) | 1.5 | (0.6) | 1.5e-4 |
| methyl ethyl ketone (MEK) | n.d. | (-) | - | (-) | n.d. | (-) | - | (-) | 9.0 | (0.3) | 2.3e-4 |
| butylacetate | 2.9 | (1.3) | 5.1e-5 | (4.3e-5) | n.d. | (-) | - | (-) | n.d. | (-) | |
| 4-acetyl-1-methylcyclohexene | n.d. | (-) | - | | 1.3 | (0.6) | 2.2e-4 | (1.05e-4) | 4.5 | (4.0) | 0.008 |
| nopinone | 4.8 | (3.2) | 0.0018 | (0.0012) | 2.9 | (2.5) | 0.0010 | (0.0009) | 10 | (4) | 0.003 |
| *Organic acids* | | | *0.071* | *(0.013)* | | | *0.024* | *(0.018)* | | | *0.02* |
| acetic acid | 2800 | (446) | 0.060 | (0.008) | 1507 | (430) | 0.020 | (0.014) | 3007 | (283) | 0.01 |
| propanoic acid | 142 | (25) | 0.0044 | (0.0008) | 84 | (26) | 0.0018 | (0.0013) | 160 | (52) | 0.001 |
| butanoic acid | 98 | (37) | 0.0046 | (0.0017) | 58 | (33) | 0.0019 | (0.0017) | 152 | (26) | 0.002 |
| isobutanoic acid | n.d. | (-) | - | (-) | n.d. | (-) | - | (-) | 32 | (14) | 2.8e-4 |
| pentanoic acid | 20 | (13) | 0.0016 | (0.0015) | 21 | (11) | 5.5e-4 | (0.9e-4) | 176 | (36) | 0.005 |

| | 29 – 30 April | | | | 22 – 25 May | | | | 24 – 26 July | |
|---|---|---|---|---|---|---|---|---|---|---|
| | Mixing ratio [ppt$_v$] | | $R_{OH}$ [s$^{-1}$] | | Mixing ratio [ppt$_v$] | | $R_{OH}$ [s$^{-1}$] | | $R$ | |
| | mean | (std) | mean | (std) | mean | (std) | mean | (std) | | mean |
| isopentanoic acid | 1.0 | (0.2) | 1.3e-6 | (2.2e-6) | 2.0 | (0.5) | 2.5e-6 | (23.4e-6) | 8.5 | (5.2) | 2.2e-4 |
| hexanoic acid | 5.9 | (2.1) | 5.2e-5 | (21.6e-5) | 9.7 | (2.8) | 5.5e-5 | (27.5e-5) | 35 | (11) | 0.001 |
| 4-methylpentanoic acid | n.d. | (-) | - | (-) | n.d. | (-) | - | (-) | n.d. | (-) | |
| heptanoic acid | n.d. | (-) | - | (-) | n.d. | (-) | - | (-) | 13 | (3) | 5.0e-4 |
| *Inorganics* | | | *1.2* | *(0.2)* | | | *1.0* | *(0.2)* | | | *1.* |
| NO | 77 | (41) | 0.012 | (0.012) | 89 | (52) | 0.013 | (0.013) | 69 | (43) | 0.01 |
| NO$_2$ | 449 | (374) | 0.14 | (0.12) | 418 | (295) | 0.11 | (0.08) | 149 | (94) | 0.03 |
| O$_3$ | 4.3e4 | (9e3) | 0.066 | (0.015) | 4.2e4 | (7.4e3) | 0.69 | (0.014) | 2.7e4 | (8.9e3) | 0.04 |
| SO$_2$ | 53 | (60) | 9.0e-4 | (13.1e-4) | 37 | (24) | 5.8e-4 | (45.4e-4) | 74 | (114) | 0.0001 |
| CO | 1.29e5 | (9e3) | 0.72 | (0.06) | 1.10e5 | (6e3) | 0.58 | (0.03) | 1.4e5 | (2e4) | 0.7 |
| CH$_4$ | 1.938e6 | (2e3) | 0.23 | (0.01) | 1.923e6 | (5e3) | 0.26 | (0.02) | 1.9e6 | (2.2e-16) | 0.2 |
| *Model OVOCs* | | | *0.19* | *(0.06)* | | | *0.20* | *(0.09)* | | | *0.6* |
| *Model inorganics* | | | *0.057* | *(0.008)* | | | *0.084* | *(0.026)* | | | *0.07* |
| *Missing* | | | *61* | *(40)* | | | *13* | *(8)* | | | *1* |
| *Total* | | | *64* | *(41)* | | | *15* | *(10)* | | | *2* |

\* quantified with $\beta$-caryophyllene calibration and an estimated reaction coefficient (1e-10 cm$^3$ s$^{-1}$)